# Impaired fast-spiking interneuron function in a genetic mouse model of depression

**Jonas-Frederic Sauer\*, Michael Strüber, Marlene Bartos\***

Physiologisches Institut I, Systemic and Cellular Neurophysiology, Albert-Ludwigs-Universität Freiburg, Freiburg, Germany

**Abstract** Rhythmic neuronal activity provides a frame for information coding by co-active cell assemblies. Abnormal brain rhythms are considered as potential pathophysiological mechanisms causing mental disease, but the underlying network defects are largely unknown. We find that mice expressing truncated *Disrupted-in-Schizophrenia 1* (Disc1), which mirror a high-prevalence genotype for human psychiatric illness, show depression-related behavior. Theta and low-gamma synchrony in the prelimbic cortex (PrlC) is impaired in Disc1 mice and inversely correlated with the extent of behavioural despair. While weak theta activity is driven by the hippocampus, disturbance of low-gamma oscillations is caused by local defects of parvalbumin (PV)-expressing fast-spiking interneurons (FS-INs). The number of FS-INs is reduced, they receive fewer excitatory inputs, and form fewer release sites on targets. Computational analysis indicates that weak excitatory input and inhibitory output of FS-INs may lead to impaired gamma oscillations. Our data link network defects with a gene mutation underlying depression in humans.

## Introduction

Psychiatric disorders not only diminish life quality of affected individuals, but also pose a substantial issue in public health because of their high prevalence in modern society. Mutations in 'risk genes' enhance the probability to develop these disorders, pointing to a strong genetic component in the etiology of mental illnesses (*Ross et al., 2006*). More than two decades ago, *DISC1* has been identified as a major genetic risk factor involved in psychiatric disorders (*St Clair et al., 1990*; *Blackwood et al., 2001*). The original discovery came from a Scottish family carrying a large c-terminal 1:11 translocation in the *DISC1* gene downstream of exon eight, which results in a c-terminal truncation of *DISC1* (*St Clair et al., 1990*). Family members who are affected by the mutation suffer from mental illness including major depression (10 cases), schizophrenia (7 cases) or bipolar disorder (1 case, *St Clair et al., 1990*). By comparison (48 cases), no major psychiatric disease was diagnosed in any of the relatives lacking *DISC1* truncation. Thus, truncation of *DISC1* constitutes one of the largest known risk factors for mental illness.

Recently, a mouse model has been developed which reproduces the human form of *DISC1* truncation (*Shen et al., 2008*). Those Disc1 mice allow to directly examine the impact of a depression- and schizophrenia-related risk gene mutation on behaviour and the activity of neuronal networks involved in the control of cognitive functions. We find that Disc1 mice show increased immobility during the tail-suspension (TST) and forced swim test broadly accepted as depression-related behavioural changes in rodents (*Porsolt et al., 1977*; *Steru et al., 1985*). This behavioural phenotype correlates with abnormalities in the synchrony of low-gamma oscillations (30–50 Hz) in the prelimbic cortex (PrlC), which have been implicated in supporting encoding of information (*Fries et al., 2007*). Moreover, we provide evidence that network malfunction is related by a profound defect of FS-INs, including reduced numbers of FS-INs as well as alterations in their synaptic connections. Thus, our study provides a correlative link between behavioural alterations in Disc1 mice and the possible underlying cellular and synaptic mechanisms.

**\*For correspondence:** jonas.
sauer@physiologie.uni-freiburg.de
(JS); marlene.bartos@physiologie.
uni-freiburg.de (MB)

**Reviewing editor**: Indira M Raman, Northwestern University, United States

**eLife digest** Our thoughts and emotions are produced and processed by complex networks of neurons inside our brains. Signals are sent from one neuron to another via chemical messengers, and pass through the neuron as an electrical signal. The electrical signals produced by a brain region often show steady rhythms, or oscillations. In the brains of many people diagnosed with certain mental disorders, such as schizophrenia and major depression, these oscillations are disrupted, but how these changes in rhythm are linked to defects in the networks of neurons behind the electrical activity is not well understood.

Studies of a family in Scotland over several decades revealed that a gene called *DISC1* was shortened in family members who had been diagnosed with mental illnesses. Recently, scientists have been able to create mice that have mutations that are equivalent to this *DISC1* mutation. It is hoped that studying the behavior and neural activity of these mutant mice could lead to a better understanding of human mental disorders.

Sauer et al. confirmed that the mutant mice showed depression-related behavior; in experiments that involved trying to escape from hopeless situations, the mutant mice gave up on their escape attempts much sooner than the normal mice. Recording the brain activity of these 'depressed' mice showed that the activity of a brain region called the prelimbic cortex was weak and disordered—very much like the brain activity seen in human depression. In particular, two types of brain activity, called theta and low-gamma oscillations, were not synchronized.

To determine precisely what causes these abnormal oscillations, Sauer et al. took brain slices from depressed mice, and then stained them with dyes that showed the circuits in the prelimbic cortex more clearly. This revealed that depressed mice had developmental defects in a specific type of inhibitory neuron called fast-spiking interneurons—there were fewer of these cells, and the neurons that were there did not have the correct number of connections to other neurons. Further investigation showed that these neurons had difficulties receiving and releasing the chemical messengers that allow neurons to communicate, and Sauer et al. thought that this might cause the low-gamma oscillation problems.

To confirm this theory, Sauer et al. created a computer model that simulated the defective interneurons. The simulations support the theory that the defects in the fast-spiking interneurons cause the abnormal low-gamma rhythms seen in depressed mice. In the future, a better understanding of the defects of inhibitory cells in *DISC1* mutants and other mouse models of mental illness might open up new avenues for targeted drug design. As the prelimbic cortex combines inputs from various other brain areas, a further challenge will be to examine whether these inputs influence the activity of the prelimbic cortex and thus contribute to depression-related behavior.

## Results

### Behavioural analysis of Disc1 mice

To investigate the effect of truncated Disc1 on potential depression- or schizophrenia-like phenotypes of Disc1 mice we conducted a comprehensive analysis covering a wide spectrum of behavioural deficits characteristic for psychiatric syndromes (*Figure 1*, refer to *Table 1* for a summary of values). We probed depression-related traits using highly validated tests for anhedonia (*LeGates et al., 2012*) and behavioural despair (*Porsolt et al., 1977*; *Steru et al., 1985*) in rodents. Disc1 mice showed no deficit in sucrose preference, which is used to quantify anhedonia (*Figure 1A*). However, in the TST and forced swim test, in which animals exhibit epochs of immobility that are thought to reflect states of behavioural despair intersected by periods of active escape, Disc1 mice showed longer periods of immobility (*Figure 1B,C*). Behavioural variability among individuals was high and therefore resulted in a moderately but significantly enhanced mean immobility by 35% in the TST and by 18% in the forced swim test of Disc1 mice (TST: p = 0.015, 22 Disc1 and 14 control mice; forced swimming: p = 0.049, 22 and 20 mice; Cohen's *d* of 0.82 and 0.65 corresponding to a strong and moderate effect size, respectively [*Table 1*]). Both genotypes reached similar movement speeds in the open field arena, indicating that high immobility in the behavioural despair tests cannot be caused by motor impairment (*Figure 1D*).

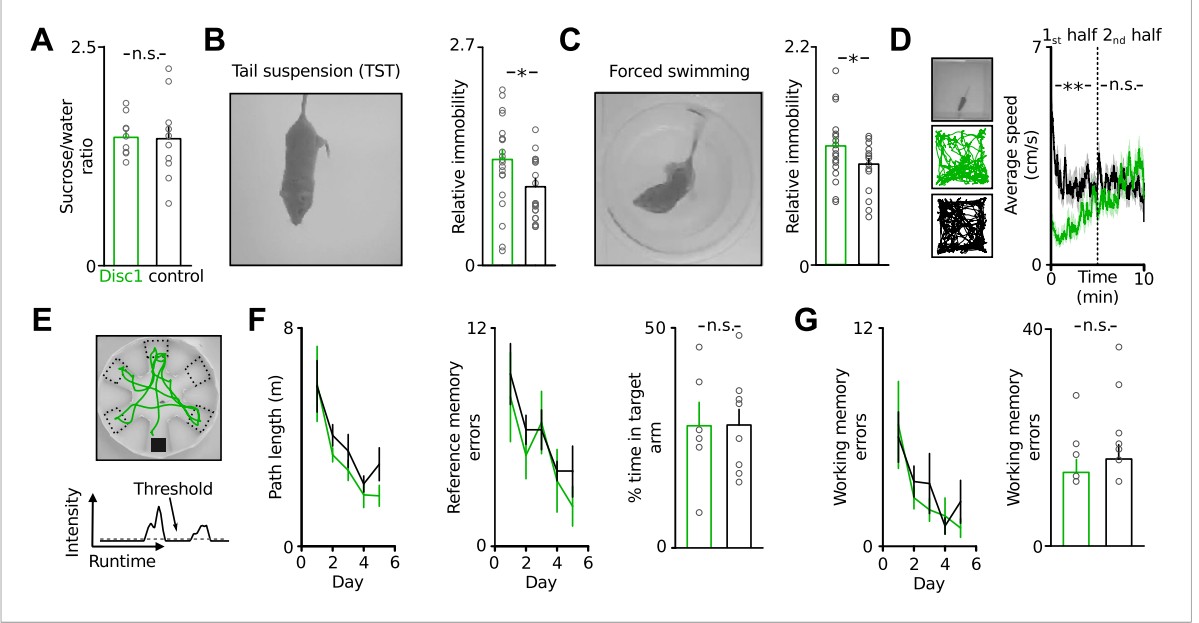

Figure 1. Disc1 mice show depression-related behavioural despair. (A) Disc1 mice show similar sucrose preference as controls (146 ± 7 vs 144 ± 15% sucrose intake, n = 10 each group). (B and C) Enhanced behavioural despair of Disc1 mice in TST (37.8 ± 3.3 vs 25.8 ± 2.9%, n = 22 Disc1, 14 control mice) and forced swim test (44.1 ± 2.6 vs 37.3 ± 2.0%, n = 22, 20). (D) Unaltered locomotion of Disc1 mice. Left, examples of the path of a Disc1 and a control mouse during a 10 min exploration period (n = 19 Disc1, 18 control mice). Right, Disc1 mice move slower during the initial phase of the task but reach similar movement speeds as controls during the later phase. (E) Radial arm water maze to probe spatial reference and working memory. The animals were released from a random start arm and had to find the hidden platform in the southern arm. Green line shows the path of one representative animal during one trial. Arm entries were detected by a threshold-crossing algorithm (bottom, N = 6, 9). (F) Path length and reference memory errors plotted against the five subsequent test days. Identical shortening of the swim path length (left) and identical number of reference memory errors (entries in wrong arms; middle) of Disc1 and control mice indicates intact spatial learning. Time spent in the target arm is identical between genotypes (n = 6, 9). (G) The number of working memory errors (re-entries in previously explored arms within a trial, 14 ± 2 vs 17 ± 3) did not depend on the genotype. *p < 0.05, **p < 0.01. Data are mean ± SEM.

The following figure supplements are available for figure 1:

Figure supplement 1. Additional assessment of working memory performance.

Figure supplement 2. Extra-dimensional rule-shifting task in the y-maze.

Figure supplement 3. Unaltered anxiety of Disc1 mice.

Figure supplement 4. Unaltered sociability of Disc1 mice.

To test for schizophrenia-associated symptoms we assessed context representation and learning (*Waters et al., 2004*) and examined spatial reference and working memory in the radial arm water maze (*Murray et al., 2011*). Both groups showed identical spatial learning and numbers of working memory errors (*Figure 1E–G*). We confirmed intact working memory of Disc1 mice in a delayed match-to-sample and a spontaneous alternation task (*Figure 1—figure supplement 1*). Furthermore, Disc1 mice could normally learn reward rules in a spatial extra-dimensional paradigm-shifting task (*Figure 1—figure supplement 2*). This test resembles features of the Wisonsin card sorting test, in which schizophrenia patients typically show deficits (*Okubo et al., 1997*). Finally, Disc1 mice had no abnormalities in anxiety or sociability (*Figure 1—figure supplements 3, 4*). Thus, Disc1 mice showed a specific phenotype broadly interpreted as depression-related behavioural despair (*Porsolt et al., 1977*; *Steru et al., 1985*).

## Synchrony of gamma oscillations is markedly reduced in Disc1 mice

PrlC integrates information from cortical and subcortical regions to exert higher-level control of behaviour including the decision to execute actions (*Yee, 2000*) and the regulation of mood

**Table 1**. Quantitative summary of cellular and synaptic properties of Disc1 and control PrlC neurons

| Parameter | Disc1 | Control | P/N | Test | Cohen's d |
|---|---|---|---|---|---|
| Sucrose preference | 1.46 ± 0.07 | 1.44 ± 0.15 | 0.901, N = 10/10 | Student's t-test | na |
| Freezing in TST (norm.) | 1.39 ± 0.12 | 1.00 ± 0.10 | 0.015, N = 22/14 | Student's t-test | 0.82 |
| Immobility in forced swim test (norm.) | 1.18 ± 0.07 | 1.00 ± 0.05 | 0.049, N = 22/20 | Student's t-test | 0.65 |
| Open field total path length (m) | 14.5 ± 0.9 | 19.3 ± 0.7 | 0.0012, N = 19/18 | Student's t-test | 1.47 |
| Open field path length (first half/second half) | 4.8 ± 0.4/8.9 ± 0.5 | 9.8 ± 0.5/9.1 ± 0.3 | $1.1*10^{-12}$, 0.682 for s half | One-way ANOVA followed by t-test | na (second half) |
| Radial arm maze reference/ working memory errors | 22 ± 5/14 ± 2 | 25 ± 2/17 ± 3 | 0.239/0.361, N = 6/9 | Mann–Whitney U test | na |
| Proportion spontaneous alternation | 0.60 ± 0.03 | 0.65 ± 0.04 | 0.361, N = 8/8 | Student's t-test | na |
| Extradim. spatial/shifted rule learning trial (y-maze) | 10 ± 1/39 ± 7 | 15 ± 3/34 ± 9 | 0.154/0.632, N = 8/7, 6/5 | Student's t-test | na |
| Time in center of open field (%) | 43.5 ± 0.9 | 37.4 ± 3.5 | 0.377, N = 19/18 | Student's t-test | na |
| 3-chamber social interaction: stranger preference | 49.4 ± 4.6 | 53.2 ± 3.3 | 0.513, N = 9/9 | Student's t-test | na |
| TST-dependent cFos increase PrLC (norm.) | 3.92 ± 1.54 | 4.78 ± 1.01 | 0.03/0.04, N = 4/3 | Mann–Whitney U test | 1.52/3.49 |
| Freezing in TST (%, electrode-implanted sample) | 52.2 ± 0.1 | 29.4 ± 0.1 | 0.004, N = 8/6 | Mann–Whitney U test | 1.82 |
| TST: low-gamma power/ amplitude ($*10^{-3}$) | 0.11 ± 0.02/0.87 ± 0.05 | 0.29 ± 0.04/1.34 ± 0.09 | 0.003/0.002, N = 8/6 | Mann–Whitney U test | 2.33/2.77 |
| TST: theta power/amplitude ($*10^{-3}$) | 0.11 ± 0.03/0.04 ± 0.004 | 0.29 ± 0.04/0.07 ± 0.004 | 0.012/0.001, N = 8/6 | Mann–Whitney U test | 1.97/3.12 |
| Home cage: low-gamma power ($*10^{-3}$) | 0.12 ± 0.02 | 0.22 ± 0.03 | 0.009, N = 8/6 | Mann–Whitney U test | 1.62 |
| Home cage: theta power ($*10^{-3}$) | 0.1 ± 0.02 | 0.18 ± 0.04 | 0.031, N = 8/6 | Mann–Whitney U test | 1.29 |
| Urethane anesthesia: low-gamma power ($*10^{-3}$) | 0.6 ± 0.2 | 1.1 ± 0.2 | 0.024, N = 11/7 | Student's t-test | 1.27 |
| TST: hippocampal theta power ($*10^{-3}$) | 0.8 ± 0.2 | 4.1 ± 1.5 | 0.009, N = 5/4 | Mann–Whitney U test | 1.71 |
| CA1-PrLC theta coherence TST/home cage | 0.52 ± 0.06/0.52 ± 0.07 | 0.52 ± 0.04/0.49 ± 0.02 | 0.972, N = 4/3 | One-way ANOVA | na |
| PV-IN count PrLC all layers (normalized) | 0.69 ± 0.08 | 1.00 ± 0.04 | 0.004, N = 9/8 | Student's t-test | 1.81 |
| PV-IN count layer 2–3/layer 5 | 3.0 ± 0.7/31.9 ± 5.3 | 6.2 ± 0.54/48.8 ± 2.52 | 0.007/0.011, N = 6/5 | Mann–Whitney U test | 2.35/1.86 |
| Somatostatin-IN count (normalized) | 1.04 ± 0.10 | 1.00 ± 0.08 | 0.392, N = 6/5 | Mann–Whitney U test | na |
| DAPI density PrLC (normalized) | 0.93 ± 0.04 | 1.00 ± 0.04 | 0.158, N = 6/5 | Mann–Whitney U test | na |
| Calbindin-IN count (normalized) | 0.75 ± 0.12 | 1.00 ± 0.05 | 0.027, N = 6/5 | Mann–Whitney U test | 1.2 |
| PV-VGAT-positive boutons PrLC (normalized) | 0.62 ± 0.09 | 1.00 ± 0.10 | 0.021, N = 6/4 | Mann–Whitney U test | 2.02 |
| FS-IN bouton density in vitro ($\mu m^{-1}$) | 0.086 ± 0.011 | 0.073 ± 0.009 | 0.388, N = 12/10* | Student's t-test | na |
| FS-IN bouton density in vivo ($\mu m^{-1}$) | 0.091 ± 0.010 | 0.093 ± 0.010 | 0.868, N = 16/20* | Student's t-test | na |

*Table 1. Continued on next page*

*Table 1. Continued*

| Parameter | | Disc1 | Control | P/N | Test | Cohen's d |
|---|---|---|---|---|---|---|
| FS-IN axon length in vitro (mm) | | 1.54 ± 0.14 | 2.02 ± 0.27 | 0.200, N = 4/4* | Mann–Whitney U test | na |
| | Amplitude (pA) | 50.8 ± 13.3 | 121.1 ± 32.1 | 0.011, N = 20/13* | Mann–Whitney U test | 0.79 |
| | rise time (ms) | 0.32 ± 0.02 | 0.32 ± 0.03 | 0.985, N = 19/13* | Mann–Whitney U test | na |
| | decay time constant (ms) | 5.92 ± 0.40 | 5.28 ± 0.28 | 0.193, N = 10/11* | Mann–Whitney U test | na |
| | onset latency (ms) | 1.01 ± 0.03 | 1.02 ± 0.05 | 0.378, N = 19/13* | Mann–Whitney U test | na |
| FS-IN-to-PC uIPSC | failure rate | 0.32 ± 0.07 | 0.08 ± 0.04 | 0.015, N = 20/13* | Mann–Whitney U test | 1.06 |
| | coefficient of variation | 0.924 ± 0.165 | 0.426 ± 0.072 | 0.008, N = 17/13* | Mann–Whitney U test | 0.92 |
| | skewness | −0.262 ± 0.135 | −0.124 ± 0.174 | 0.503, N = 17/13* | Mann–Whitney U test | na |
| | paired-pulse ratio 20/50 ms | 0.84 ± 0.08/0.85 ± 0.08 | 0.89 ± 0.09/0.82 ± 0.03 | 0.836/0.937, N = 7/6, 6/6 | Mann–Whitney U test | na |
| | multiple-pulse 50 Hz 10th | 0.58 ± 0.06 | 0.54 ± 0.05 | 0.671, N = 8/7* | Student's t-test | na |
| Connection probability (%) | | 34.6 | 12.8 | 5.3*10–29, N = 78/148* | Chi2 test | na |
| | Nr | 4 ± 2 | 10 ± 2 | 0.036, N = 14/11* | Mann–Whitney U test | 0.87 |
| Binomial fitting | Qr | 27.6 ± 2.6 | 28.5 ± 3.5 | 0.831, N = 14/11* | Student's t-test | na |
| | Pr | 0.45 ± 0.07 | 0.58 ± 0.06 | 0.178, N = 14/11* | Student's t-test | na |
| | amplitude (pA) | 30.7 ± 2.3 | 28.7 ± 3.4 | 0.34, N = 15/9* | Mann–Whitney U test | na |
| spEPSCs on FS-INs | frequency (Hz) | 6.0 ± 1.0 | 9.5 ± 1.1 | 0.022, N = 15/9* | Student's t-test | 1.1 |
| | coefficient of variation | 0.75 ± 0.04 | 0.75 ± 0.05 | 0.596, N = 15/9* | Student's t-test | na |
| | amplitude (pA) | 17.9 ± 1.1 | 18.4 ± 3.0 | 0.202, N = 11/13* | Mann–Whitney U test | na |
| spEPSC on PCs | | | | | | |
| | frequency (Hz) | 3.1 ± 0.4 | 2.6 ± 0.4 | 0.246, N = 11/13* | Mann–Whitney U test | na |
| | amplitude (pA) | 20.1 ± 1.1 | 21.9 ± 1.9 | 0.388, N = 24/15* | Student's t-test | na |
| mIPSC on PCs | | | | | | |
| | frequency (Hz) | 0.74 ± 0.14 | 1.18 ± 0.23 | 0.025, N = 24/15* | Mann–Whitney U test | 0.57 |

N indicates number of animals except for: *N indicates number of cells, #N indicates number of axons.

(*Covington et al., 2010*), both of which are impaired in depression (*Elliott et al., 1997*). We therefore hypothesized that PrlC dysfunction may be involved in the emergence of behavioural despair of Disc1 mice. By using antibody labelling against the immediate early gene cFos as a marker of neurons which underwent enhanced activity, we found that TST increased the number of cFos-positive cells in the PrlC compared to baseline in the home cage in both Disc1 and control mice, indicating that the PrlC is involved in controlling behavioural despair (*Figure 2A*). The elevation of cFos-positive cells by TST occurred in both, Disc1 as well as control mice, suggesting that no major differences in PrlC activation upon exposure to TST may exist between genotypes (*Figure 2A*).

In humans and rodents activity in the PrlC becomes synchronized during various behavioural states (*Uhlhaas and Singer, 2010*; *Roux et al., 2012*). We therefore speculated that Disc1 truncation might cause changes in rhythmic activity patterns rather than gross activity levels in the PrlC. To test for this possibility, we recorded local field potentials (LFPs) in the PrlC of behaving mice (*Figure 2B*). Longer freeze times of Disc1 mice were apparent in the electrode-implanted sample (8 Disc1 and 6 control mice, p = 0.0041; *Figure 2B*) similar to non-implanted Disc1 mice. During TST, Disc1 mice showed reduced normalized power and amplitude in the theta (6–12 Hz, p = 0.012 and p = 0.001, respectively) and low-gamma (30–50 Hz, p = 0.003 and p = 0.002, respectively) but not high gamma band (80–100 Hz, p = 0.11 and p = 0.239, respectively; *Figure 2C–E*). Oscillations were similarly impaired during home

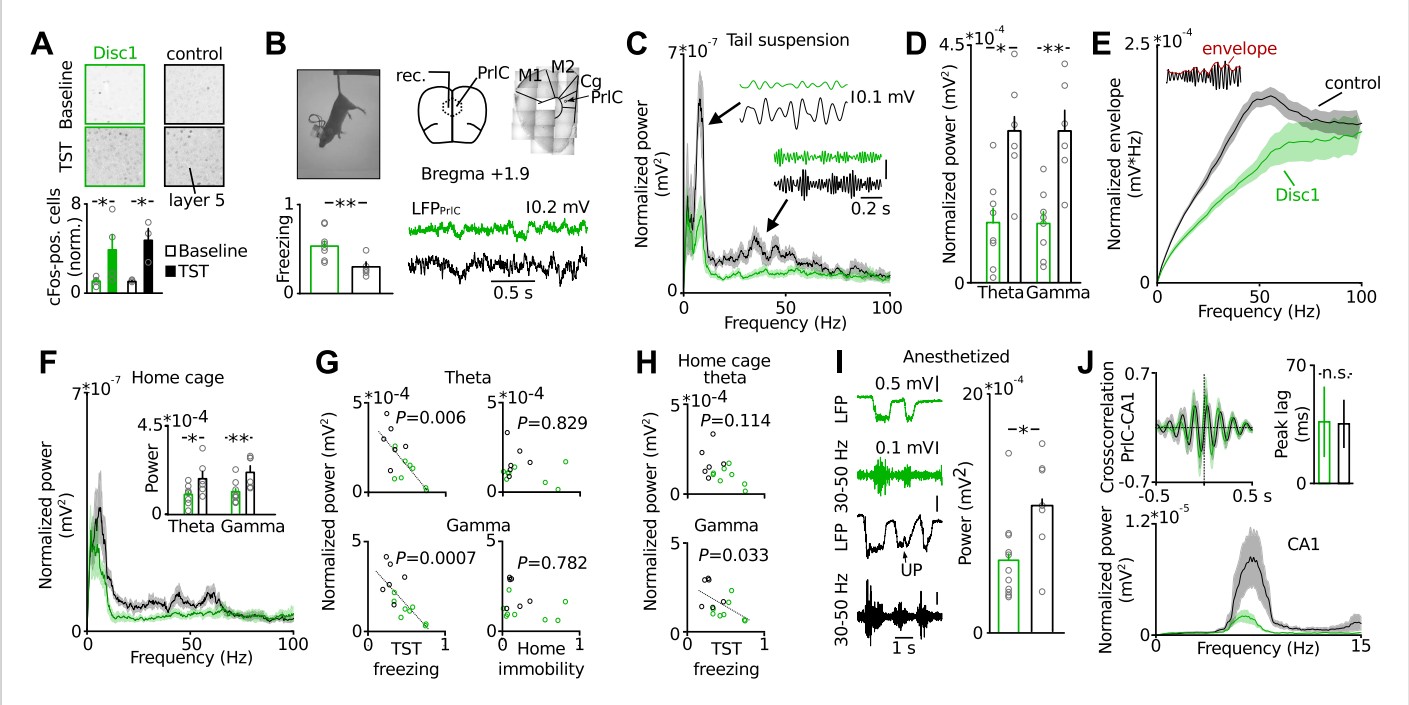

**Figure 2**. Behavioural despair of Disc1 mice correlates with impairment in theta and low-gamma oscillations in the PrlC. (**A**) TST activates cFos in PrlC independent from genotype (fold increase Disc1: 3.92 ± 1.54, n = 4; control: 4.78 ± 1.07, n = 3). (**B**) LFP recording during TST. Enhanced freezing of Disc1 mice is preserved in the electrode-implanted cohort (52.2 ± 5.8 vs 29.4 ± 4.0, n = 8, 6). M1,2: motor cortex, Cg: cingulate cortex. (**C** and **D**) Reduced power of Disc1 mice in the theta (0.11 ± 0.03 vs 0.29 ± 0.04 mV$^2$*10$^{-3}$) and low-gamma band (0.11 ± 0.02 vs 0.29 ± 0.04 mV$^2$*10$^{-3}$, n = 8, 6). Insets: filtered traces. (**E**) Oscillation amplitudes over frequency. (**F**) Oscillatory defects are observed in the home cage (theta: 0.10 ± 0.02 vs 0.18 ± 0.04 mV$^2$*10$^{-3}$, gamma: 0.12 ± 0.02 vs 0.22 ± 0.03 mV$^2$*10$^{-3}$, n = 8, 6). (**G**) Theta and low-gamma power correlate with TST freezing duration (theta: r = −0.6923, p = 0.0061; low-gamma: r = −0.79, p = 0.0008) but not with home cage immobility (r = −0.029, r = −0.222). Black lines: linear fits. (**H**) Home cage low-gamma but not theta can predict TST freezing (gamma: r = −0.569, theta: r = −0.440). (**I**) Low-gamma activity in Disc1 PrlC is impaired during UP-states in anesthesia (0.6 ± 0.1 vs 1.1 ± 0.2 mV$^2$*10$^{-3}$, n = 11, 7). (**J**) Top, cross-correlation of LFP simultaneously recorded in hippocampus and PrlC suggests that theta oscillations are driven by hippocampus (peak lag: 36.5 ± 20.9 vs 35.3 ± 14.3 ms, n = 5, 4). Bottom, hippocampal theta power is impaired in Disc1 mice (0.87 ± 0.23 vs 4.14 ± 1.54 mV$^2$*10$^{-3}$, n = 4, 3, p = 0.01). *p < 0.05, **p < 0.01. Data are mean ± SEM, circles are individual mice.

The following figure supplements are available for figure 2:

**Figure supplement 1**. Unchanged hippocampal-prefrontal theta coherence.

**Figure supplement 2**. Low-gamma defect in the PrlC of Disc1 mice does not depend on the behavioral state during TST.

cage exploration (*Figure 2F*). These observations were independent of the behavioral state and present during movement and passive coping (*Figure 2—figure supplement 1*). When data from Disc1 and control mice were pooled, theta and low-gamma power linearly correlated with freeze duration in TST (theta$_{TST}$ p = 0.0061 and low-gamma$_{TST}$ p = 0.0007, respectively; *Figure 2G*, left) but not with immobility in the home cage (theta$_{baseline}$ p = 0.829 and low-gamma$_{baseline}$ p = 0.782, respectively; *Figure 2G*, right), showing that reduced synchrony of theta and low-gamma oscillations impairs TST-specific cortical processing rather than alterations in general locomotion. Moreover, low-gamma but not theta power in the home cage could significantly predict TST freeze duration when Disc1 and control data were pooled (p = 0.033, p = 0.114, respectively; *Figure 2H*), indicating that local low-gamma power correlates with defects of the intrinsic function of the prefrontal network irrespective of the animal's behaviour. Recordings of spontaneous low-gamma oscillations during UP states under anaesthesia (*Hasenstaub et al., 2005*) confirmed the independence of low-gamma power from behaviour-dependent brain state (*Figure 2I*). Jointly these data suggest that defective theta$_{TST}$ and low-gamma$_{TST+baseline}$ are correlated with behavioural despair of Disc1 mice.

We next examined the mechanisms underlying oscillatory impairments in Disc1 PrlC. Prefrontal theta oscillations are driven by the hippocampus (*Siapas et al., 2005*; *Sigurdsson et al., 2010*) whereas gamma activity patterns are generated by synaptic interactions between GABAergic FS-INs and glutamatergic PCs in local neuronal networks (*Atallah and Scanziani, 2009*; *Tiesinga and Sejnowski, 2009*). Consistent with the hippocampal drive of theta oscillations to the PrlC, cross-correlation analysis of theta-filtered signals in simultaneous LFP recordings from dorsal CA1 and PrlC revealed a ~30 ms peak time lag in both Disc1 and control mice (*Figure 2J*). In agreement with the intact working memory of Disc1 mice, for which high synchrony of theta oscillations between hippocampus and prefrontal cortex is required (*Jones and Wilson, 2005*; *Siapas et al., 2005*; *Sigurdsson et al., 2010*), coherence in the theta band was comparable between genotypes (*Figure 2—figure supplement 2*). However, theta power was markedly reduced in CA1 of Disc1 mice (*Figure 2J*), suggesting that the prefrontal theta power deficit may be caused by a theta dysfunction in the hippocampus.

## Number of parvalbumin-positive interneurons is reduced in the prefrontal cortex of Disc1 mice

To obtain deeper insight into the pathophysiology of Disc1-associated behavioural despair, we next focussed on the mechanisms underlying impaired low-gamma oscillations in the PrlC because human depression patients show reduced low-gamma activity in frontal regions (*Liu et al., 2012*). Our cFos labelling suggested that TST directly activated the PrlC network (*Figure 2A*) and that local PrlC mechanisms may contribute to the TST phenotype of Disc1 mice. The PrlC of Disc1 mice contained significantly fewer PV-positive INs (~40% reduction, p = 0.0037, 9 Disc1 and 8 control mice, *Figure 3A*, *Figure 3—figure supplement 1*). A similar reduction in PV-positive cells was observed in CA1 (~40%; p = 0.022) but not in the ventro-orbital cortex (p = 0.375; *Figure 3—figure supplement 1*). PV-positive cells of both genotypes expressed Disc1 (*Figure 3—figure supplement 2*). In contrast, the number of somatostatin-expressing INs (p = 0.392) and total cell density (DAPI area, p = 0.158) were unchanged (*Figure 3A,B*). Studies on schizophrenia patients suggested that PV-expression might be down-regulated in FS-INs (*Hashimoto et al., 2003*). However, detection of PV immunoreactivity in electrophysiologically identified FS-INs in PrlC slices did not depend on the genotype (Disc1: 9/16 cells; control: 10/19 cells; *Figure 3C*, *Figure 3—figure supplement 3*). Moreover, the number of INs expressing calbindin, a marker for FS-INs partially coexpressed with PV (*Markram et al., 2004*), was reduced in the Disc1 PrlC in vivo (~25% reduction, p = 0.027, 6 Disc1 and 5 control mice, *Figure 3D*), supporting our conclusion of reduced PV-cell quantity rather than PV content of FS-INs. Finally, the frequency of miniature IPSCs (mIPSCs) recorded in PCs was significantly reduced in the PrlC of Disc1 mice, consistent with a loss of PV-positive cells (p = 0.025, 24 Disc1 and 15 control cells; *Figure 2E*). These data further suggested a lack of mechanisms compensating for the reduced PV cell population.

The loss of FS-INs was paired with an equal reduction in the number of their PV/VGAT-coexpressing terminals (~40% reduction, p = 0.021, 6 Disc1 and 4 control mice, *Figure 3F*). Three-dimensional reconstructions of FS-INs revealed no difference in bouton density on FS-IN axons (*Figure 3G*). Similarly, we detected comparable bouton densities on PV-positive axons in the PrlC of both genotypes in vivo (*Figure 3G*) and indistinguishable axon lengths from reconstructed cells in vitro (*Figure 3—figure supplement 4*). Thus, the amount of PV-positive FS-INs and their synapses is reduced by ~40% in the Disc1 PrlC.

## Reduced inhibitory output and excitatory input signaling of fast-spiking parvalbumin-positive interneurons in the prefrontal cortex of Disc1 mice

To test whether altered synaptic transmission of local FS-INs might contribute to the low-gamma defect, we recorded from layer 5 FS-INs and PCs in acute prefrontal slices (*Figure 4*). Paired recordings from synaptically connected FS-INs and postsynaptic PCs revealed a strong reduction in the amplitude of unitary inhibitory postsynaptic currents (uIPSCs) in Disc1 PrlC (~60% reduction, p = 0.012, 20 and 13 pairs, respectively; *Figure 4A*). This decline in synaptic inhibition was not caused by a reduced connection probability with distance among communicating partners because inter-somatic distances between pre- and postsynaptic cells were identical (<60 μm; see 'Material and methods'; *Figure 4—figure supplement 1*).

To determine which of the synaptic parameters, number of release sites ($N_r$), quantal size ($Q_r$) and release probability ($P_r$), may contribute to the reduction in uIPSC size, we used multiple

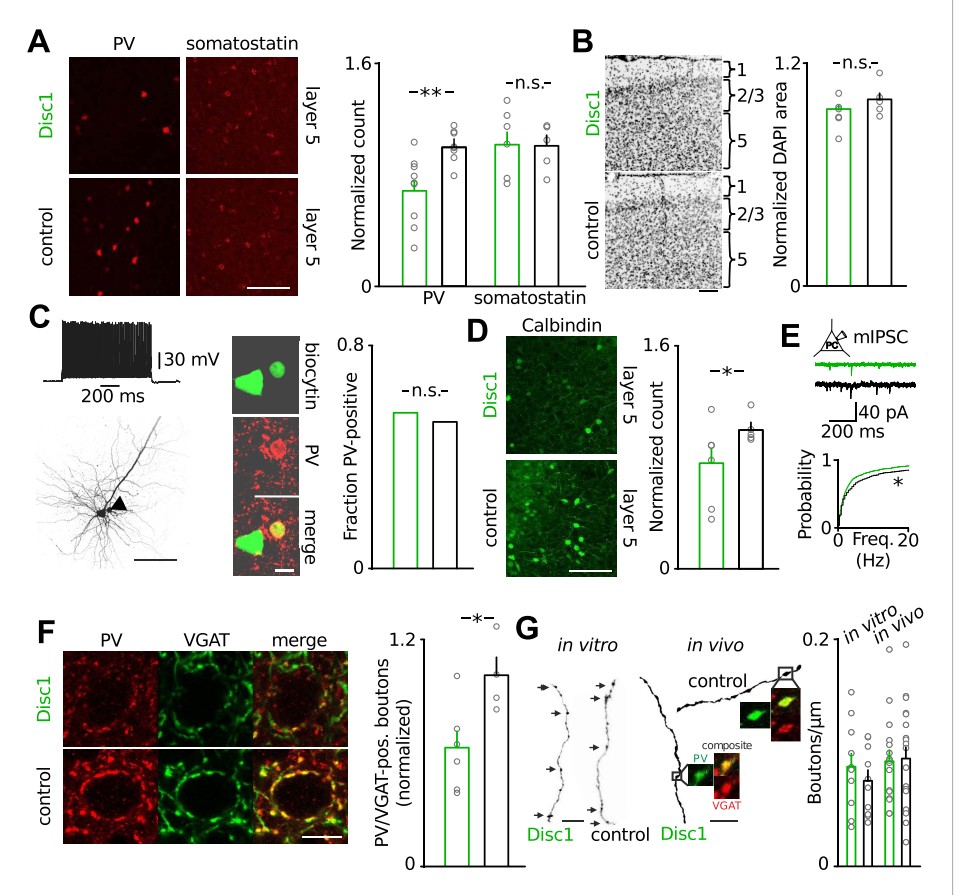

**Figure 3**. Loss of FS-INs and their output synapses in the Disc1 PrlC. (**A**) Reduction of PV but not somatostatin-positive INs in the PrlC of Disc1 mice (normalized count PV: 0.69 ± 0.08 vs 1.00 ± 0.04, $n$ = 9 Disc1, 8 control mice, p = 0.0037; somatostatin: 1.04 ± 0.10 vs 1.00 ± 0.08, $n$ = 6, 5, p = 392). (**B**) Total cell density quantified from DAPI area is unchanged (normalized density: 0.93 ± 0.04 vs 1.00 ± 0.04, $n$ = 6, 5 mice, p = 0.158). (**C**) FS-INs of Disc1 and control mice express PV ($n$ = 25, 29 cells). (**D**) Expression of the FS-IN marker calbindin is reduced in Disc1 PrlC (0.75 ± 0.12 vs 1.00 ± 0.05, $n$ = 6, 5 mice, p = 0.027). (**E**) Frequency of mIPSCs recorded in PCs was significantly reduced in the Disc1 PrlC (0.73 ± 0.14 vs 1.18 ± 0.23 Hz, $n$ = 24, 5 cells, p = 0.025). (**F**) Fewer PV-VGAT-coexpressing boutons in Disc1 mice (normalized count 0.62 ± 0.09 vs 1.00 ± 0.10, $n$ = 6, 4 mice, p = 0.021). (**G**) Identical bouton density of intracellularly labelled FS-INs in vitro (0.09 ± 0.01 vs 0.07 ± 0.01 μm$^{-1}$, $n$ = 12, 10, p = 0.388) and PV-positive cells in vivo (0.09 ± 0.01 vs 0.09 ± 0.01 μm$^{-1}$, $n$ = 16, 20, p = 0.868). *p < 0.05, **p < 0.01. Scale bars: **A**, **B**, **C** (left), **D**: 100 μm, **E**: 25 μm, **C** (right): 10 μm. Data are mean ± SEM, circles individual mice or cells (**C** and **F**).

The following figure supplements are available for figure 3:

**Figure supplement 1**. PV-positive INs are lost throughout cortical layers in the Disc1 PrlC.

**Figure supplement 2**. PV-positive INs express Disc 1.

**Figure supplement 3**. Electrophysiological characteristics of FS-INs.

**Figure supplement 4**. Morphological characteristics of FS-INs.

probability-compound binomial analysis (*Kraushaar and Jonas, 2000*) (*Figure 4B,C*). *Nr* but not *Qr* or *Pr* was reduced in Disc1 pairs by ~60% (p = 0.036, p = 0.831, p = 0.178, 14 and 11 pairs, respectively; *Figure 4C*). Bootstrapping demonstrated that errors in the parameter estimation were similar to previous reports (*Kraushaar and Jonas, 2000*) (*Figure 4—figure supplement 2*). Failure rate and coefficient of variation of uIPSCs were higher in Disc1 pairs (p = 0.015 and p = 0.008, respectively;

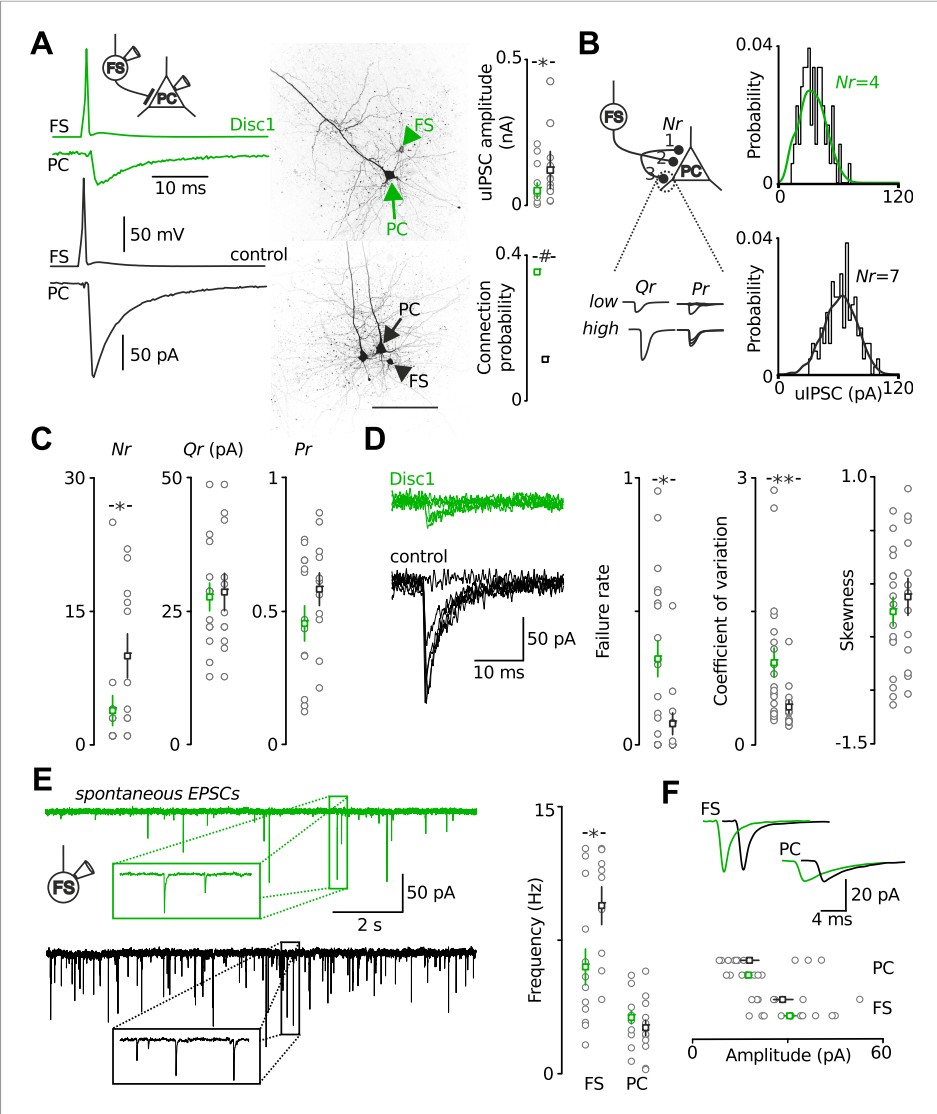

**Figure 4**. Output and input signalling of FS-INs are impaired in the Disc1 PrlC. (**A**) Paired recordings of FS-INs and PCs revealed a reduction in uIPSC amplitude in the PrlC of Disc1 mice (50.8 ± 13.3 vs 121.1 ± 32.1, n = 20, 13 pairs, p = 0.012) but enhanced connection probability (35.4 vs 11.5%, n = 65, 122 simultaneous recordings, p < 0.001). Scale, 100 μm. Middle, confocal images of the pairs. (**B**) Amplitude distributions of uIPSCs of a Disc1 and control pair. Lines represent best fit results obtained by multiple probability compound binomial analysis. (**C**) Nr but not Qr or Pr are reduced at Disc1 FS-IN synapses (Nr: 4 ± 2 vs 10 ± 2, p = 0.036; Qr: 27.6 ± 2.6 vs 28.5 ± 3.5, p = 0.831; Pr: 0.45 ± 0.07 vs 0.58 ± 0.06, p = 0.340, n = 14, 11 pairs). (**D**) Left, superimposed single traces of a Disc1 and control pair. Failure rate, coefficient of variation, and skewness of uIPSCs support reduced Nr (0.32 ± 0.07 vs 0.08 ± 0.04, p = 0.015, 0.92 ± 0.17 vs 0.43 ± 0.07, p = 0.008, −0.262 ± 0.135 vs −0.124 ± 0.174, p = 0.503, respectively, n = 17–20, 13 pairs). (**E**) Reduced frequency of spEPSCs in Disc1 FS-INs (6.0 ± 1.0 vs 9.5 ± 1.1 Hz, p = 0.022, n = 15, 9) but not PCs (3.1 ± 0.4 vs 2.6 ± 0.4, p = 0.246, n = 11, 13). (**F**) Unchanged spEPSC amplitudes (FS-INs: 30.7 ± 2.3 vs 28.7 ± 3.4 pA, p = 0.340, n = 15, 9 cells; PCs: 17.9 ± 1.1 vs 18.4 ± 3.0 pA, p = 0.246, n = 11, 13 paris). *p < 0.05, #p < 0.001. Data are mean ± SEM, circles individual cells.

The following figure supplements are available for figure 4:

**Figure supplement 1**. Recording depths and axonal distance between pairs of synaptically connected neurons are similar in Disc1 and control FS-IN to PC recordings.

*Figure 4. Continued*

**Figure supplement 2**. Bootstrapping analysis reveals small errors in the estimation of synaptic parameters with binomial fitting.
**Figure supplement 3**. Identical dynamic and kinetic properties of uIPSCs at FS-IN output synapses.
**Figure supplement 4**. Unaltered Q at Disc1 FS-IN-to-PC synapses.
**Figure supplement 5**. Similar fluctuation of spEPSCs at Disc1 and control FS-IN inputs.

*Figure 4—figure supplement 1*), whereas the skewness was unchanged (p = 0.503, *Figure 4D*), confirming a change in *Nr* rather than *Pr* (*Kerr et al., 2008*). Paired-pulse behaviour and kinetic properties of uIPSCs did not depend on the genotype, further excluding altered *Pr* or somatodendritic synapse location, respectively (*Figure 4—figure supplement 3*). Amplitudes of quantal IPSCs recorded in the presence of extracellular 5.5 mM strontium were not significantly different between genotypes (4 and 5 pairs; p = 0.195). Moreover, mIPSCs had similar mean size in PCs located in the PrlC of Disc1 and control mice, further confirming similar *Qr* (24 and 15 cells, p = 0.388; *Figure 4—figure supplement 4*). Thus, Disc1 FS-INs form ~60% fewer release sites per target PC, resulting in an according reduction of unitary inhibitory strength. How can the contradiction between similar numbers of axonal release sites per FS-IN but fewer synaptic contacts per FS-IN-to-PC connection in Disc1 PrlC be reconciled? Interestingly, connection probability defined as the probability to record from connected FS-IN-to-PC pairs was ~threefold higher in Disc1 mice (Disc1: 35.4%, control: 11.5%, p < 0.001; *Figure 4A*), suggesting that redistribution of release sites at the expense of individual connection strength might contribute to low-gamma defects.

Recruitment of FS-INs by local excitatory collaterals is an important requirement for the generation of gamma oscillations (*Tiesinga and Sejnowski, 2009*). We therefore examined FS-IN excitation by glutamatergic synapses (*Figure 4E,F*). The frequency of spontaneous excitatory postsynaptic current (spEPSC) was strongly reduced (p = 0.022, 15 Disc1 and 9 control cells). In contrast the mean amplitude and coefficient of variation of spEPSCs were unchanged (p = 0.34 and p = 0.596, respectively; *Figure 4E, F*, *Figure 4—figure supplement 5*), suggesting reduced PC-to-FS-IN connectivity rather than changes in *Nr*, *Qr* or *Pr*. SpEPSC frequency in PCs was unaffected (*Figure 4E,F*). Thus, synaptic excitation particularly of FS-INs is impaired and may contribute to low-gamma defects in Disc1 PrlC.

## Computational analysis reveals the parameters underlying reduced gamma synchrony in a Disc1 neuronal network model

In the Disc1 PrlC fewer FS-INs redistribute their weaker outputs to a higher number of PCs and receive fewer glutamatergic inputs. To address whether these alterations influence the synchrony of low-gamma oscillations, we designed computational neuronal network models with synaptically connected FS-INs and PCs and compared scenarios with experimentally-driven synaptic properties and connectivities from Disc1 and control prefrontal cortices (*Wang and Buzsáki, 1996*) (*Figure 5*, *Table 1*, *Table 2*). Both network models generated synchronous low-gamma activity patterns (*Figure 5B*). These oscillations were generated by a recurrent PC → FS-IN → PC network over a broad range of excitatory drives provided to both FS-INs and PCs (*Figure 5C*, *Figure 5—figure supplement 1*), in agreement with current theories on the generation of gamma rhythms in cortical networks (*Tiesinga and Sejnowski, 2009*). Consistent with gamma oscillations in prefrontal areas of rodents (*Massi et al., 2012*) and monkeys (*Wilson et al., 1994*), INs discharged at higher rates than PCs (mean f$_{AP}$; *Figure 5B*). Spike histograms as well as LFP analogs demonstrated high synchrony of low-gamma activity in the control network model (*Figure 5B,C*; black). In contrast, reduced synchrony of low-gamma emerged in the Disc1 circuit (*Figure 5B,C*; green; *Figure 5—figure supplement 1*). Precise timing of PC activity was proposed an important requirement for information processing (*Uhlhaas and Singer, 2010*). Cross-correlation analysis of FS-IN and PC discharges and quantification of PC spike times in relation to FS-IN activity revealed that spike timing fidelity of PCs was high in the control but strongly reduced in the Disc 1 network model (*Figure 5C*; *Figure 5—figure supplement 1*). These findings were robust over a wide range of excitatory regimes (*Figure 5—figure supplement 1*).

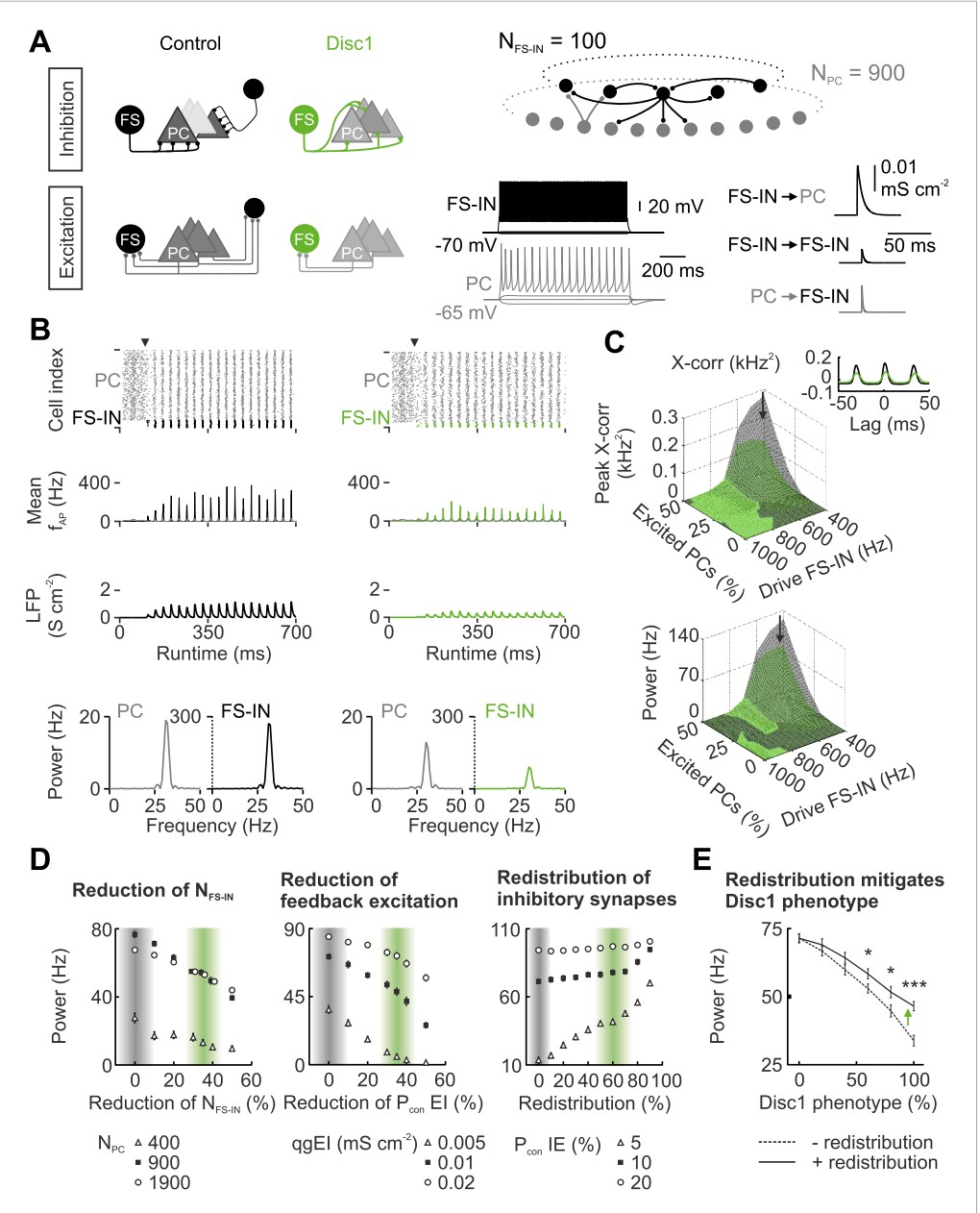

**Figure 5**. Disc1-mediated circuit changes impair low-gamma power in a network model. (**A**) Schematic of network structure, cellular and synaptic properties of the control (black) and Disc1 circuit (green). (**B**) After synapses are enabled (arrowhead), networks synchronize in the low-gamma range. From top to bottom: Raster plots representing action potentials; binned spike frequencies; LFPs; power of firing rates. (**C**) Cross-correlation of PC and FS-IN activity and power under different regimes of Poisson-distributed excitatory drives. Arrows point to simulation in (**B**). Inset, PC-FS-IN spike cross-correlogram. (**D**) Effects of separate Disc1-induced circuit changes on low-gamma power. Left, effect of fewer FS-INs at different PC population sizes. Middle, reduced feedback excitation expressed as connection probability ($P_{con}$) for different E-I quantal conductances (qg). Right, redistribution of inhibitory synapses (reduction of release sites/connection with proportional increase in $P_{con}$ I-E for networks with different mean connection probabilities). Shaded bars indicate control and Disc1 parameters. Filled squares represent the default network. (**E**) Low-gamma power depends on the strength of the Disc1 phenotype. Dashed line: reduced $N_{FS-IN}$ and $P_{con}$ EI. Continuous line: additional redistribution of inhibitory synapses. *p < 0.05, #p < 0.001.

The following figure supplement is available for figure 5:

**Figure supplement 1**. Characteristics of Disc1 and control low-gamma oscillations at different excitatory drives.

**Table 2**. Summary of default paramters defining intrinsic and synaptic properties in network models

| Synapse | $P_{con}$ (%) | $P_{con}$ Disc1 (%) | $Sd_{con}$ (IN–IN spacings) | gq (mS cm$^{-2}$) | Nr | Nr Disc1 | $\tau_{rise}$ (ms) | $\tau_{decay}$ (ms) | $E_{syn}$ (mV) | $g_{gap}$ (nS) |
|---|---|---|---|---|---|---|---|---|---|---|
| IN-IN | 20 | 50 | 30 | 0.005 | 10 | 4 | 0.15 | 2 | −60 | 0.01 |
| IN-PC | 10 | 25 | 30 | 0.02 | 10 | 4 | 0.2 | 5.5 | −60 | – |
| PC-IN | 10 | 6.5 | 20 | 0.01 | 5 | 5 | 0.1 | 1 | 0 | – |
| exc. Drive | | | | 0.05 | 1 | 1 | 0.1 | 2 | 0 | – |

Abbreviations: $P_{con}$, probability of synaptic connections; $SD_{con}$, standard deviation in connection probability expressed as cell-to-cell distances; qg, quantal conductance; Nr, number of release sites; $\tau_{rise}$ and $\tau_{decay}$, rise and decay time constant; $E_{syn}$, synaptic equilibrium potential; $g_{gap}$, electrical coupling.

Finaly, we isolated the identified changes in Disc1 PrlC (reduced FS-IN number, diminished local feedback excitation, and redistribution of inhibitory contacts) and examined their individual influence on low-gamma power (*Figure 5D,E*). Reducing FS-IN number and their feedback excitation to experimentally defined Disc1 values resulted in a cumulative decline of low-gamma power (*Figure 5D, E*). In contrast, redistribution of FS-IN output synapses, which reproduced reduced unitary inhibitory strength and enhanced connectivity, improved network synchrony but failed to lift it to control levels (*Figure 5E*; p < 0.001). Taken together, these data propose that loss of FS-INs, their reduced recruitment and diminished inhibitory output strength jointly result in an impairment of low-gamma synchrony in the PrlC.

## Discussion

Our results demonstrate that a mutation in a high risk gene for depression, the *DISC1* truncation correlates with reduced synchrony of theta and low-gamma oscillations in the PrlC. Notably and in line with the intact working memory of Disc1 mice (*Sigurdsson et al., 2010*), phase-locking between hippocampal and prefrontal regions was unchanged. In contrast, our results suggest that reduced low-gamma synchrony in the PrlC may contribute to enhanced immobility of Disc1 mice, interpreted as depression-related behaviour (*Porsolt et al., 1977*; *Steru et al., 1985*), and the extent of synchrony reduction predicted the magnitude of the phenotype. The mean Disc1 phenotype differed mildly but significantly from controls which can be largely explained by the high inter-individual variability in behaviour. We identified impaired synaptic excitatory input and inhibitory output of PV-FS-INs in the Disc1 PrlC as a strong candidate mechanism underlying low-gamma defects. Thus, truncation of *DISC1* in human patients may contribute to the development of depression by affecting anatomical and physiological properties of prefrontal PV-FS-INs.

Our conclusions fit to the key role of FS-INs in the generation of fast network oscillations in cortical networks (*Cardin et al., 2009*). Single cell recordings during spontaneous gamma oscillations in vivo (*Massi et al., 2012*; *Pernia-Andrade and Jonas, 2014*) and during pharmacologically induced gamma activity patterns in vitro (*Hajos et al., 2004*) revealed that strength of perisomatic inhibition and timed synaptic feedback excitation of FS-INs are key parameters setting gamma synchrony in healthy cortical networks, including the PrlC (*Goldman-Rakic, 1995*) and the hippocampus (*Hajos et al., 2004*). Furthermore, optophysiological activation or silencing of PV-IN populations result in the enhancement or suppression of gamma power in the prefrontal cortex in vivo, respectively (*Sohal et al., 2009*). Similarly, recruitment of FS-INs by activating PC assemblies increases gamma power in the somatosensory cortex (*Cardin et al., 2009*). Thus, the observed reduced excitatory input to FS-INs, which may cause diminished recruitment of these cells, as well as the impaired synaptic output of FS-INs will ultimately result in a loss of gamma synchrony. Gamma synchrony is also sensitive against changes in strength of gap coupling among FS-INs (*Bartos et al., 2002*). Whether this synaptic property is altered in Disc1 mice would need further investigations. Our conclusions fit also to recent investigations in cortical slice preparations from mouse models for psychiatric disorders caused by mutations of genes encoding the Lysophosphatidic acid 1 receptor (*Cunningham et al., 2006*), neuregulin-1 or ErbB4 receptor (*Fisahn et al., 2009*). Impairment of pharmacologically induced gamma activity in these studies correlated with a marked loss of cortical PV-expressing cells. Although the mechanisms underlying the loss of synchronous gamma oscillations were not examined in these investigations, they indicate that excitation-inhibition imbalance may be the pathophysiological mechanism.

PV-cells are expressed throughout the brain raising the question of specificity of the observed Disc1 effects for the prefrontal cortex. The number of PV-positive cells was also reduced in CA1 but unaltered in the ventro-orbital cortex, suggesting that PV cell defects similar to the ones identified in the PrlC occur in some but not necessarily all brain regions containing PV cells. Thus, in depth investigations are required to understand brain area-specific differences in Disc1 effects.

What cellular mechanisms might explain the structural and functional reorganization of the Disc1 PrlC? Studies of Disc1 protein interactions indicate a central role in developmental processes such as morphological differentiation and neuronal migration (*Ozeki et al., 2003*; *Duan et al., 2007*). Recent examinations established a direct link between Disc1 and axon growth. Disc1 knockout resulted in reduced axon elongation in cultured hippocampal cells (*Shinoda et al., 2007*). Neurotrophin-3 signaling via phosphorylation of extracellular signal related kinase 1/2 (ERK-1/2) is a crucial regulator of axon development, and Disc1 knockdown has been shown to abolish this phosphorylation (*Shinoda et al., 2007*). Recent findings in the hippocampus further suggest a role for Disc1 in axonal path-finding. The axons of dentate gyrus granule cells, the mossy fibers, were redistributed from their normal location in strata lucidum and oriens towards the PC layer of CA3 in a mouse model expressing truncated Disc1 (*Faulkner et al., 2008*). Disc1 is highly expressed in the developing brain (*Ozeki et al., 2003*; *Schurov et al., 2004*), highlighting its likely role in neuronal maturation and network wiring. Although no reports specifically for GABAergic axons exist, these results suggest that the effects of *DISC1* truncation are not PC-specific but may also affect FS-IN axons during development. Finally, Disc1 is required for the migration of IN precursors from the ganglionic eminences to the cortex (*Steinecke et al., 2012*). Therefore, truncated *DISC1* might cause a migratory block of FS-INs, which could lead to the observed reduced PV cell numbers in the adult prefrontal cortex of this study.

Given the limited efficacy of current anti-depressive medication, deciphering the mechanisms underlying network defects caused by mutations of *DISC1* or other 'risk genes' will be a crucial step towards new treatment options.

## Materials and methods

### Behaviour

All animal experimentation was in agreement with national legislation (approved by the *Regierungspräsidium Freiburg*). During behavioural tests, adult Disc1 mice (*Shen et al., 2008*) (>5 weeks) were housed with 2–5 animals per cage. Animals were accustomed to handling in daily sessions for at least two days prior to experimentation. For TST, the tail of the mice was fixed with tape to a horizontal bar at ~25 cm height. Movement was recorded with an IP camera. All mice were tested once. The forced swim test was performed in a 2 l glass beaker filled with 1 l tap water. Both tests lasted 6 min. For open field analysis, the mice were placed in a 30 × 30 cm arena and videotaped for 10 min. Two regions of interest of identical area (center, periphery) were defined. The percentage time spent in both regions and the total distance travelled were quantified. To test for anhedonia, mice were housed individually with two drinking bottles per cage, one containing 1% sucrose solution, the other tap water. Liquid consumption from both bottles was measured after 48 hr. Prior to this test, mice were kept on a two-bottle paradigm for 2–5 days. The position of the sucrose containing flask was chosen randomly for each cage. Spatial reference and spatial working memory were tested in a radial 6-arm water maze in a bathing pool (120 cm diameter). The water (19–21°C) was whitened with non-toxic tempera colour. Mice were released from a random start arm and allowed to find the hidden platform within 1.5 min. After this time window mice were guided to the platform by the experimenter. Animals were allowed to rest on the platform for 15 s. Four runs per day were performed for 5 days with fixed target and random starting arm locations. After the last run a probe trial was conducted with the platform removed from the maze. Arm entry detection and tracking of movement was performed automatically with custom-made ImageJ routines based on MTrack2 and Python routines. Spatial reference memory errors were defined as entries in non-target arms and spatial working memory errors as re-entries in previously explored non-target arms within a trial. A match-to-place task was performed in a 3-arm water maze with a start arm, a target arm with hidden platform, and a non-target arm. Each mouse first underwent a 30 s extinction trial with only start arm and non-target arm open. After a 30 s inter-trial interval, the mouse was put back in the start arm, this time with both arms accessible (sample trial). Animals were allowed to find the platform within 1.5 min and were guided to the platform in case they failed to perform the task. Mice rested on the platform for 15 s upon

arrival. Finally, after another 30 s inter-trial interval, the match trial task was conducted similarly to the sample trial. A trial was defined as correct if the mouse first entered the target arm in the match trial (3 runs per day for 4 days). The extradimensional paradigm-shifting test was carried out in a Y-maze. Mice were food-restricted for 5 days. During this time, they were trained to search for food reward available at the end of both target arms upon release from the start arm. From day 6 onward, mice received food reward in the right arm ('right correct' task). During all tasks, both target arms were randomly illuminated with LEDs. When mice reached the learning criterion (10 subsequent correct runs or 1 error in 12 runs), the reward rule was switched to 'light on—correct arm'. Learning was measured in 10 subsequent runs and from learning curves computed with the Learning Analysis toolbox (*Smith et al., 2004*). Social behaviour was measured in a 3-chamber social interaction arena composed of two side chambers and one central chamber (30 × 19 cm each). Both side chambers contained a wire pencil cup. In the first habituation run, mice explore the arena. In the second run, a stranger mouse was placed in one randomly chosen cup (individual run duration 10 min). The total travel distance and time spent in each compartment were quantified. For all tests, animals were randomly chosen and test apparatuses, except the water maze, were cleaned with 70% ethanol between animals.

## Surgery for in vivo electrophysiology

Adult animals were anesthetized with isoflurane (induction 3%, maintenance 1–2%) for chronic implantations or urethane (injected intraperitoneally; 2 g/kg urethane in saline) for recordings in fully anesthetized conditions. Mice were fixed in a stereotaxic frame (Kopf Instruments) and received $O_2$ through a mouthpiece throughout the procedure. Body temperature was kept stable with a heating pad set to 38°C. Stereotaxic coordinates for the PrlC were (from bregma): anterior-posterior: +1.9 to +2 mm, medio-lateral: 2–2.25 mm, and 1.6–1.7 mm forward at 45° from brain surface for anesthetized recordings or +1.9, 0.7–0.8 and 1.9 from brain surface at 10–15° for chronic implantations. CA1 coordinates were from bregma: −2 mm, 1.5 mm and 1.5 mm (PC layer of PC layer/stratum oriens border). Coordinates were determined with a mouse brain atlas. LFPs were recorded in anaesthetized mice with either glass pipettes filled with physiological saline (resistance 0.5–3 MΩ) or tungsten microelectrodes (80 µm tip diameter, HEKA; 5 kHz sampling frequency). LFPs in freely moving mice were recorded with an implanted teflon-insulated platinum or stainless-steel wire (125 µm diameter) fixed with superglue/dental cement mixture. A reference electrode was placed over the parietal cortex or cerebellum. Buprenorphine (0.02–0.03 ml) was injected subcutaneously at the end of the surgery.

The recording site of all animals was identified by perfusing them intracardially (see below) with 4% paraformaldehyd (PFA) after recordings. Brains were sectioned and stained with cresyl violet. In a subset of experiments, animals were sacrificed by decapitation under deep urethane anaesthesia and brains were fixed in 4% PFA for 2–10 days. Horizontal slices were cut (300 µm thickness), washed in phosphate-buffered saline (PBS) and embedded in Mowiol. Recording sites were identified using a light microscope (Zeiss 2FS Plus). Only experiments with identified recording sites in the PrlC or the CA1 pyramidal cell layer to stratum oriens border were used for data analysis.

## In vivo electrophysiology and data analysis

LFPs were measured under anaesthesia with an EPC10 USB amplifier (HEKA) in current clamp mode. A wireless amplifier system (W4, Multichannel Systems) was used for LFPs recordings during behaviour >2 days after surgery (1 kHz sampling frequency). LFPs were analyzed with open-source MATLAB routines (MathWorks; www.chronux.com). Power spectral density and coherence were computed with the 'multi-taper' functions of the Chronux toolbox using nine data tapers. Power spectra and envelopes were corrected for the 1/f decline in power over frequency by multiplying power at each frequency by frequency ('normalized power'). This normalization did not affect differences in power between genotypes (data not shown). For cross-correlation analysis MATLAB's *xcorr* function was used. Gamma envelopes were extracted from the absolute of the Hilbert transform calculated with custom made Python routines.

## In vitro electrophysiology and data analysis

Frontal PrlC slices were cut from 3-to-4 week-old animals as described before (*Sauer and Bartos, 2010*). In paired recordings action potentials were induced in the presynaptic FS-IN with short duration current injection (2 ms) while the postsynaptic PC was monitored in voltage-clamp with a holding potential ($V_{hold}$) set to −70 mV. Only neurons with intersomatic distances <60 µm were

targeted for paired recordings. Recording temperature was 32–34°C. Neurons were judged not connected when repeated (>20) trials of action potentials in the presynaptic cell did not elicit a postsynaptic signal in the target neuron as monitored at high-resolution settings with an oscilloscope. Only recordings with an access resistance <25 MΩ were accepted for analysis. Spontaneous EPSCs (spEPSCs) were pharmacologically isolated by bath application of SR95531 (5 µM). Amplitude and time course of mean spEPSCs were analyzed with a threshold-crossing algorithm using Python. Tetrodotoxin (0.5 µM) and kynurenic acid (4 mM) were bath-applied to isolate mIPSCs.

Unbinned uIPSC amplitude distributions were fit with a multiple probability-compound binomial analysis model of release consisting of the sum of Nr Gaussian functions representing 1 to Nr individual independent release sites, thereby following the procedure of (*Kraushaar and Jonas, 2000*). Free parameters for the procedure were Qr, standard deviation of Qr (SDQr), Pr and Nr with initial boundaries set to 10–50, 3–30, 0.1–0.9 and 1–31, respectively. The step sizes for iteration were 4, 5.4, 0.1 and 3, respectively. For each parameter combination a probability density function was created as a sum of Gaussians with means at Qr*i and width SDQr*i for i in range 1 to Nr. Each Gaussian was multiplied with the binomial probability of release at i release sites with a given Pr (*Kraushaar and Jonas, 2000*). Failures were included as the binomial probability of release at zero sites and set as point zero of the compound probability density function. The best fit of the data was determined with the maximum-likelihood-estimation procedure. To receive confidence intervals of the estimated parameters we performed bootstrap analysis (*Kraushaar and Jonas, 2000*). For bootstrap analysis the original amplitudes of an experiment were loaded and bootstrap datasets were created from the n data points as n random picks with replacement. Binomial fitting was then performed on the bootstrap data with maximum-likelihood-estimation. The whole procedure was repeated 100- times for each experiment. Quantal uIPSCs were recorded in pairs with extracellular $CaCl_2$ replaced with $SrCl_2$ (5.5 mM). Asynchronous release was triggered under these conditions by evoking action potential trains (10 pulses, 50 Hz) in the presynaptic neuron. Asynchronously released quanta were detected up to 400 ms after the train.

## Immunohistochemistry

Mice were deeply anesthetized by brief isoflurane exposure followed by intraperitoneal injection of pentobarbital (0.2 ml of 15 mg/ml solution in water) or urethane (2 g/kg in physiological saline). Surgery was only commenced after pain reflexes had been abolished. Mice were transcardially perfused with PBS for ~1–2 min, then with 4% PFA for ~13–30 min (~3–8 ml/min). Brain was removed and stored in PBS overnight and in a subset of experiments in 4% PFA. Horizontal slices (50 µm) were permeabilized in PBS and 0.4% Triton X-100 (30 min at room temperature), blocked in PBS and 0.2% Triton X-100 and 4% normal goat serum (NGS) for 30 min (room temperature). Primary antibodies were applied overnight at 4°C in PBS and 0.1% Triton X-100 and 2% NGS. Slices were washed three times in PBS with 1% NGS (10 min), incubated in secondary antibody solution (PBS, 1.5% NGS; 2–3 hr at room temperature) and subsequently washed in PBS (2 × 10 min). DAPI was applied for 5 min (1:1000 in PBS). After final washing steps in PBS (3 × 10 min) slices were embedded in Mowiol. Incubation with secondary antibody alone gave no unspecific staining (data not shown). Antibody-labelling was visualized with a confocal microscope (Zeiss LSM510 or 710). The primary antibodies used were: mouse-anti-PV (Swant; 1:1000), mouse-anti-calbindin (Swant; 1:1000), rabbit-anti-SOM (Peninsula Laboratories; 1:1000), rabbit-anti-VGAT (Synaptic Systems; 1:1000), rabbit-anti-Disc1 (Sigma–Aldrich; 1:1000), and rabbit-anti-cFos (Calbiochem, 1:2000). The secondary antibodies used were: Cy3-goat-anti-rabbit (Jackson Immunoresearch or Dianova; 1:1000), AlexaFluor647-goat-anti-mouse (Invitrogen; 1:1000) and AlexaFluor488-goat-anti-mouse (Invitrogen; 1:1000).

Cell bodies were counted from maximum intensity projections of z-stacks taken with a 10× objective or from epifluorescence images taken at 5× magnification. To assess PV/VGAT double-positive boutons, colocalized structures were visually identified in 100 × 100 µm regions in 40× single-z plane images within layer 5 of the PrlC. Cells and boutons were counted manually without knowing the genotype. Data were compared with an automated approach in which colocalization was quantified from the same images with custom-made routines written in ImageJ's macro language. Detection thresholds were set to both channels independently with the ImageJ 'triangle' method until 5, 10, 25, 50 or 70 brightest percent of pixels remained. Both channels were multiplied with each other to reveal the fraction of colocalized areas.

As s control, the same analysis was performed with VGAT images rotated by 90˚ to determine random colocalizations which were subtracted from colocalizations. Automated analysis gave results that were comparable to manual counting (data not shown). To define bouton density in FS-INs, boutons of intracellularly labelled cells were visually identified as brighter and thicker spots in the biocytin labelled axon. Axon segments (60–130 µm length) were chosen pseudorandomly from the labelled neuron. In vivo, boutons of individual PV-expressing axons (27–124 µm length) were identified in confocal z-stacks as structures colocalizing VGAT. For in vitro and in vivo analyses the length of the traced axon segment was determined in 3D with ImageJ's 'simple neurite tracer'.

## In identification and reconstruction

During whole-cell recordings, 1 s long hyperpolarizing and depolarizing current injections were applied (step size 100 pA; range −100–600 pA). INs were identified as FS when the adaptation coefficient was >0.6, determined as the ratio between the first and the last inter-spike-interval. During recordings, cells were filled with 0.2% biocytin. To visualize recorded neurons brain slices were fixed overnight in 4% PFA (4˚C), then washed in PBS (1, 10, 15, 15 min). Blocking was done for 60 min in PBS and 10% NGS. Primary antibody (mouse-anti-PV or rabbit-anti-PV, Swant, 1:1000) was incubated in PBS, 0.3% Triton X-100 and 5% NGS (24 hr at room temperature). The slices were washed again in PBS (1, 10, 15, 15 min) and transferred to secondary antibody solution composed of PBS, 0.3% Triton X-100, 3% NGS and streptavidin conjugated to AlexaFluor647 or 488 (1:500, 24 hr at 4˚C). Slices were washed in PBS (1, 10, 15, 15 min) and embedded in Mowiol. FS-INs were selected for reconstruction if the signal-to-noise ratio allowed a clear visualization of the axon. A high-resolution confocal image stack was taken with a 40× oil immersion objective (NA 1.4) at optimal resolution settings. Semi-automated 3-D reconstruction was performed with the Simple Neurite Tracer plugin of ImageJ. Skeletons of the axon were extracted and analyzed with Lmeasure (http://cng.gmu.edu:8080/Lm/). Total axon length and branch point number were normalized to the total volume of the stack.

## Network simulations

Networks of FS-INs and PCs were implemented as conductance-based single compartment models in NEURON 7.2. Three types of chemical connections were modelled: I-I, I-E and E-I, with a distance-dependent connection probability ($P_{con}$) following a Gaussian function and a mean $P_{con}$ based on our experimental observations (*Table 2*). Synaptic events were modelled as $N_r$ quantal conductance changes with an exponential rise and decay ($\tau_{rise}$ and $\tau_{decay}$) and a quantal peak conductance $qg$ obtained from our measured data (*Table 2*). Excitatory drive to the network was modelled as irregular trains of Poisson-distributed excitatory postsynaptic conductances (*Table 2*) with varying frequencies (range: 0.4–1 kHz). All presented data are averages of 20 individual simulation runs.

The neuronal network model represents a local circuit of the cortex. Networks of FS-INs (N = 100 for control and N = 65 for Disc1 networks) and PCs (default N = 900; 400–1900 in *Figure 5D*) were arranged on two concentric circles to represent the columnar organization (*Compte et al., 2000*) with minimal distances between two neighbouring cells of 50 µm. FS-INs and PCs were equipped with Hodgkin-Huxley-type conductances to reproduce the FS phenotype in INs (*Wang and Buzsáki, 1996*) and regular-spiking in PCs (*Hemond et al., 2008*). Synaptic connections were formed randomly following a distance-dependent Gaussian profile (see *Table 2*). FS-INs were electrically coupled to four of their nearest eight neighbour INs using a coupling conductance of 0.01 nS (*Bartos et al., 2002*). Strength and distribution of electrical coupling among FS-INs have not been experimentally examined in Disc1 and control mice and kept the same in both models (see *Table 2*). Mutual E–E synapses among PCs were excluded for simplicity of network design and interpretation. Events were triggered after the presynaptic action potential following a latency which consisted of a constant part (the release phase; 0.5 ms) plus a distance-dependent part (the action potential conduction phase; distance in IN–IN spacing × 0.05 ms; action potential conduction velocity 0.25 ms$^{-1}$; [*Bartos et al., 2002*]). For a single simulation run, connections were formed randomly with a random number of quantal contacts picked from the range $N_r \pm 50\%$. All cells had random initial membrane potentials (range: −70 to −60 mV) and began to receive their excitatory drive at random onset times $0 < t \leq 50$ ms. At $t = 100$ ms, all synapses were switched on. When changing the excitatory drive to the FS-IN population, the mean input frequency onto every FS-IN was altered. When changing the excitatory drive to the PC population, the mean excitatory drive on single PCs remained constant but the percentage of cells receiving that drive was varied (*Figure 5B–E*; *Figure 5—figure supplement 1*).

During the simulation, spike times and the sum of all unitary inhibitory conductances (LFP analog) were recorded and the mean firing rate histograms with 1 ms time bin were calculated (*Figure 5B*). As a measure for the strength of oscillatory activity in the network, LFP analogs and mean firing rate histograms were recorded between $400 < t \leq 700$ ms and were subjected to power spectral density analysis using MATLAB's *periodogram* algorithm with 1 Hz frequency resolution. The maximum of the resulting power spectrum indicated the prominent oscillation frequency and the power at this maximum plus the two adjacent frequencies (range: $\pm 1$ Hz) were used to quantify the mean power of the oscillation (*Figure 5B–E*). To determine time lag histograms of PC spikes, we first identified the peaks of FS-IN activity in every gamma cycle and then calculated for every PC spike the time lag to its closest FS-IN peak (*Figure 5—figure supplement 1*). We described the resulting time lag distribution with a single measure: *PC sync* = probability of a single PC spike per gamma cycle/variance ($\sigma^2$) of time lags. Using this definition, *PC sync* describes the precision of PC spikes in relation to IN activity and PC time locking to the ongoing gamma oscillation. To directly quantify the correlation between FS-IN and PC activity, mean firing rate histograms were obtained for both cell types and a cross-correlation (*X-corr*) analysis was performed using MATLAB's *xcorr* function (*Figure 5C*; *Figure 5—figure supplement 1*). Maximal cross-correlation was obtained from the peak of the resulting cross-correlogram (*Figure 5—figure supplement 1*). For default parameter settings see *Table 2*.

## Statistical testing

Statistical significance was tested with a two-tailed Student's *t*-test, Mann–Whitney U-test or signed rank test. One-way ANOVA was used for multiple comparisons. For nominally scaled data, a $\chi^2$ test or Fisher's exact test was used. Data are expressed as mean $\pm$ SEM. To measure of effect size of behavioural differences between Disc1 and control mice we used Cohen's *d*. Custom-made analysis tools are contained in the Source Code file analysiscodes.txt.

## Acknowledgements

We thank D St Clair, G Riedel, and S Shen for kindly providing Disc1 mice, U Häussler for advice with electrode implantation, K Winterhalter, M Northemann, and K Semmler for technical assistance. This work was supported by the Scottish Northern Research Partnership (JS), the Spemann Graduate School for Biology and Medicine (MS), the Lichtenberg Professorship Award (MB), the DFG (BA 1582/2-1, MB), the Schram Foundation (T287/21797/2011, MB), and Brain-Links Brain-Tools, Cluster of Excellence funded by the Deutsche Forschungsgemeinschaft EXC 1086 (MB).

## Additional information

### Competing interests
MB: Reviewing editor, *eLife*. The other authors declare that no competing interests exist.

### Funding

| Funder | Grant reference | Author |
| --- | --- | --- |
| University of Aberdeen | Scottish Northern Research Partnership | Jonas-Frederic Sauer |
| Albert-Ludwigs-Universität Freiburg (University of Freiburg) | Spemann Graduate School for Biology and Medicine | Michael Strüber |
| Volkswagen Foundation | Lichtenberg Professorship Award | Marlene Bartos |
| Schramm Foundation | T287/21797/2011 | Marlene Bartos |
| Deutsche Forschungsgemeinschaft (DFG) | Brain-Links Brain-Tools | Marlene Bartos |
| Deutsche Forschungsgemeinschaft (DFG) | BA 1582/2-1 | Marlene Bartos |

| Funder | Grant reference | Author |
| --- | --- | --- |
| Deutsche Forschungsgemeinschaft (DFG) | Cluster of Excellence | Marlene Bartos |

The funders had no role in study design, data collection and interpretation, or the decision to submit the work for publication.

### Author contributions

J-FS, Conception and design, Acquisition of data, Analysis and interpretation of data, Drafting or revising the article; MS, Acquisition of data, Analysis and interpretation of data, Drafting or revising the article; MB, Conception and design, Drafting or revising the article

### Ethics

Animal experimentation: All in vivo and in vitro examinations were performed in agreement with national legislation and institutional regulations (license nr: G-13/25; X-10/18S) approved by the 'Regierungspräsidium' Freiburg.

# Additional files

### Supplementary file

• Source code 1. Custom-made analysis tools are contained in the source code file analysiscodes.txt.

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
