## [Decision Letter]

Thank you for sending your work entitled “Impaired fast-spiking interneuron function in a genetic mouse model of depression” for consideration at *eLife*. Your article has been favorably evaluated by Eve Marder (Senior editor), a Reviewing editor, and 3 reviewers.

The Reviewing editor and the other reviewers discussed their comments before we reached this decision, and the Reviewing editor has assembled the following comments to help you prepare a revised submission.

The reviewers were overall enthusiastic about the questions addressed and the approach taken, which one reviewer described as “simply outstanding” and “superb”. The critiques centered on three major areas, summarized as follows:

1) All reviewers noted that the manuscript will be improved by a clearer distinction between causation and correlation as well as acknowledgement that the correlations seen (especially in the context of the behavioral studies) do not imply causation. As one reviewer noted, this is an issue since the existing literature already shows that Disc-1 may have pleiotropic effects in the CNS. Another reviewer drew attention to phrases of the sort “indicating that PrlC is involved controlling behavioral despair,” which imply that causality has been established. The reviewer pointed out that the only way to justify such a claim would be to specifically induce a change in PrlC and demonstrate a defect in behavioral despair in wildtype mice and then to correct the defect in a mutant mouse with a PrlC specific manipulation. These concerns regarding correlation and causation should be possible to address with some careful rewriting.

2) Another concern had to do with the extent to which the changes in PrlC were unique. The proposal is made in the Discussion that DISC1 is involved in axonal path-finding and neuronal migration. As one reviewer expressed it, because of the role played by PV fast spiking interneurons in generating gamma oscillations, one would expect similar deficits in other cortical areas, and a less specific, maybe more serious phenotype in DISC1 mice. Please indicate whether there is any reason to assume that the impairment of gamma is specific to the PrlC and/or whether other areas were examined, and please consider the issue of whether PrlC is expected to be exclusively affected in the Discussion.

3) The statistical significance of the behavioral changes was acknowledged, but their biological significance was questioned. Some discussion to help readers understand how/whether to interpret the small changes biologically would improve the manuscript.

4) There were also some concerns regarding the model-dependent quantal analysis. Additional analyses, alternative approaches, or more extensive justification might help address the reviewers' concerns on this point. Some suggestions are included in the reviewers' comments below.

Specific comments are included below to help inform your revision. Some include expansions of the four major points listed above, and some are minor issues that should be addressed in the revision.

1) The main concern is that the reasoning is very correlative and the causality is generally indirectly inferred.

A) Correlation oscillation power/behavior: Disc-1 show more immobility and less theta/gamma power, so a significant correlation is already expected in immobility vs power graphs. From the graphs it is a bit difficult to convince oneself that there is linear correlation within each genotype immobility/oscillation within each genotype for recordings in the home cage or during the TST. Since immobility may be a separate cortical state, it would be helpful to assess separately the power in active and passive states of the mice. There may not be enough data point at this stage to discriminate an independent difference of immobility/oscillation between the two genotypes from a proportional relation immobility/oscillation for WT and Disc-1 mice (as suggested by the authors).

B) Correlation oscillation/PV interneurons: the model data provide encouraging evidence for a causal role of the multiple PV defects, but this is not demonstrated. Optogenetic stimulations targeted to the PrL in Disc-1 mice might provide evidence in that direction (and even show a decrease in immobility in TST).

2) The statistical significance of the data needs to be re-assessed. The rationale for mixing parametric and non-parametric tests is unclear. Why is the score in the TST tested with a non-parametric test, while the FST uses a parametric test? Is the effect still significant with the Mann-Whitney test? Also, the TST-dependent cFos increase is tested on very small numbers (4 vs 3) for which the Mann-Whitney cannot detect significance anyway.

3) Many behavioral tests demonstrate no change in behavior of DISC1 mutants relative to wildtype animals. The only statistically significant finding is in the immobility scores in forced swim and tail suspension with p values of 0.039 and 0.049. Moreover, the sizes of the differences are very small, despite reaching significance. Such small differences are unlikely to be biologically significant. I would either remove or seriously downplay the behavioral analysis. The electrophysiological analysis can stand alone without the behavioral section.

4) The quantal analysis shown in Figure 4 is, I think, a poor way to analyze the data. Model dependent quantal analysis of the type shown allows for wide variation of parameters with little difference in the goodness of fit. Nevertheless, the authors do back up the analyses with some bootstrapping models. There are very clear changes in failure rates and coefficient of variation, which are unambiguous alterations. Some additional simple analysis with Strontium to reveal the unitary current size would allow for calculation of NP.

5) Since connection probability decreases as distance between soma increases for FS to Pyr cell pairs (Packer and Yuste, 2011), the authors should report the distance between soma of connected and unconnected pairs. An additional plot showing the relationship of distance between connected pairs and failure rate would also be informative.

6) Using anatomy (Figure 3) and physiology (Figure 4), the authors conclude that single FS terminals onto pyr cells are of the same strength but fewer in number. Does this result in a shift away from input from PV-INs onto Pyr cells, while other inhibitory inputs compensate for the loss of inhibition, or are total levels of inhibition received by pyr cells decreased? Examining mIPSCs and/or evoked IPSCs onto pyramidal cells would address this issue. This is not a necessary experiment to make the points in the paper, but would assist in constraining the models shown at the end of the paper.

7) We find sentences like “We find that DISC1 mice show depression-related behavioral despair” inappropriate. This is a particular interpretation of a simple increase the immobility of an animal. This is better suited for the Discussion and should be phrased as one particular interpretation of a subtle behavioral difference.

---

## [Author Response]

*1) All reviewers noted that the manuscript will be improved by a clearer distinction between causation and correlation as well as acknowledgement that the correlations seen (especially in the context of the behavioral studies) do not imply causation. As one reviewer noted, this is an issue since the existing literature already shows that Disc-1 may have pleiotropic effects in the CNS. Another reviewer drew attention to phrases of the sort* “*indicating that PrlC is involved controlling behavioral despair,” which imply that causality has been established. The reviewer pointed out that the only way to justify such a claim would be to specifically induce a change in PrlC and demonstrate a defect in behavioral despair in wildtype mice and then to correct the defect in a mutant mouse with a PrlC specific manipulation. These concerns regarding correlation and causation should be possible to address with some careful rewriting*.

We fully agree with this important criticism of the editor and the reviewers and made the required changes by rewording the text throughout the Results and Discussion sections.

We were highly motivated to follow the suggestion to examine whether the observed reduction in the synaptic inhibitory output strength provided by PV-cells directly relates to the reduced power of network oscillations in the prefrontal cortex and the diminished mobility in Disc1 mice. We therefore injected recombinant adeno-associated viruses (rAAVs) encoding halorhodopsin (NpHR) and the red fluorescent protein tdTomato (rAAV-NpHR-tdT) as a marker bilaterally in the prefrontal cortices of PV-Cre mice in which Cre-combinase is only expressed in fast-spiking PV-expressing interneurons. Thus, NpHR-tdT is only expressed in Cre-recombinase-expressing PV-cells. We successfully tested this approach in the hippocampus of PV-Cre mice in a previous set of experiments. After rAAV injection we implanted optical fibers for the activation of NpHR by light (565 nm) with the aim to silence fast-spiking PV-interneurons. Three weeks after virus expression we aimed to ask whether PV-cell silencing can reproduce the observed reduced PV-interneuron-mediated inhibition in the prefrontal cortex as well as reduced mobility of mice. However, we were unexpectedly confronted with several technical problems, which resulted in a time delay of our revision for which we are very sorry. The major problem was that the expression of NpHR was not homogeneous in the PV-cell population as expected from our earlier experiments. Due to the already caused time delay we decided not to repeat this set of experiments. Thus, all concerns regarding correlation and causation were addressed by careful rewriting of the manuscript.

*2) Another concern had to do with the extent to which the changes in PrlC were unique. The proposal is made in the Discussion that DISC1 is involved in axonal path-finding and neuronal migration. As one reviewer expressed it, because of the role played by PV fast spiking interneurons in generating gamma oscillations, one would expect similar deficits in other cortical areas, and a less specific, maybe more serious phenotype in DISC1 mice. Please indicate whether there is any reason to assume that the impairment of gamma is specific to the PrlC and/or whether other areas were examined, and please consider the issue of whether PrlC is expected to be exclusively affected in the Discussion*.

We thank the editors for this important criticism. To address this question we performed additional quantitative analysis on the number of PV-expressing cells in the hippocampal area CA1 and the ventro-orbital cortex (vOC). We found a similar reduction in PV-positive cells by ∼30% in CA1 (P=0.022; 7 Disc1 and 6 control mice) but not in the vOC (P=0.375). These data are now shown in Figure 3—figure supplement 1 and in the Results section under the subsection headed “Number of parvalbumin-positive interneurons is reduced in the prefrontal cortex of 129 Disc1 mice”. These data suggest that the observed cellular and synaptic defects may be also evident in other brain areas such as the hippocampus to a similar extent as observed in the PrlC but not necessarily in all brain regions. We therefore conclude that in-depth investigations are required to understand whether Disc1 effects in the PrlC are unique or more generally applicable to other cortical regions. This is discussed in the third paragraph of the Discussion section.

*3) The statistical significance of the behavioral changes was acknowledged, but their biological significance was questioned. Some discussion to help readers understand how/whether to interpret the small changes biologically would improve the manuscript*.

The variability in behaviour among individuals was high and resulted therefore in a moderate but significant enhancement in the mean immobility by ∼40% in the tail-suspension (TST) and by ∼18% in the forced swim test (FST) in Disc1 mice. We feel that a ∼1/3^rd^ change in spite of the high inter-individual variability in the TST is not a weak or subtile effect. To further strengthen this point, we computed Cohen's *d*, a measure of effect size (Lakens 2013; Front Psychol 4:863). Cohen's *d* of TST and FST were 0.82 and 0.65, respectively, which corresponds to a strong and intermediate effect size, respectively. This is mentioned in the first paragraph of the Results section, and included in Table 1. Moreover, our study shows an enormous specificity of the Disc1 mutation on the behavioural level. We would therefore prefer to keep the behavioural analysis in the manuscript. However, we follow the reviewers’ request and toned down the behavioural impact of the Disc1 mutation in the Results and Discussion.

*4) There were some concerns regarding the model-dependent quantal analysis. Additional analyses, alternative approaches, or more extensive justification might help address the reviewers' concerns on this point. Some suggestions are included in the reviewers' comments below*.

To improve the quantal analysis we performed two new sets of experiments. First, we determined the amplitude of miniature IPSCs (mIPSCs) recorded in 0.5 µM TTX in PCs of the PrlC. The rationale behind this experiment is the fact that most mIPSCs originate from PV-INs, which form the largest group of IN types in the neocortex ([23]; Nat Rev Neurosci 5:793-807). There was no difference in the amplitude of mIPSCs between both genotypes (24 Disc1 *vs* 15 control PCs; *P*=0.388). Indeed, with a mean size of ∼21 pA of both groups, it was in a similar range as *Qr* obtained with multiple probability-compound binomial model (∼28 pA), supporting the reliability of the analysis. These data are shown in the new Figure 4—figure supplement 4 of the revised manuscript and included in the second paragraph of the subsection headed “Reduced inhibitory output and excitatory input signaling of fast-spiking parvalbumin-positive interneurons in the prefrontal cortex of Disc1 mice”. Second, we followed the proposal of the reviewers and performed new-paired recordings between presynaptic fast-spiking interneurons and postsynaptic PCs in the presence of 5.5 mM strontium (SrCl_2_). The amplitude of quantal IPSCs measured under these conditions did not significantly differ among genotypes (Disc1: 10.0 ± 0.8 pA in Disc1, 4 pairs; control: 12.0 ± .1.5 pA; 5 pairs; *p*=0.195). We noted that the mean IPSC amplitudes measured in SrCl_2_ were smaller than the mean amplitudes of mIPSCs or the predictions from our probability compound binomial model. We used the equation *A*=*N***p***q* to calculate *N* for each mean IPSC amplitude (*A*) from individual paired recordings using the quantal size (*q*) obtained in SrCl_2_ recordings over a wide range of *p* values (0.2-1). This analysis showed that there is always a significantly smaller *N* in Disc1 compared to control pairs (*p*<0.05). These data are now included in Figure 4—figure supplement 4 and in the second paragraph of the subsection headed “Reduced inhibitory output and excitatory input signaling of fast-spiking parvalbumin-positive interneurons in the prefrontal cortex of Disc1 mice”.

*Specific comments are included below to help inform your revision. Some include expansions of the four major points listed above, and some are minor issues that should be addressed in the revision*.

*1) The main concern is that the reasoning is very correlative and the causality is generally indirectly inferred*.

We fully agree with the reviewer and toned down our statements throughout the manuscript (see statement to the editors’ comments, point 1).

*A) Correlation oscillation power/behavior: Disc-1 show more immobility and less theta/gamma power, so a significant correlation is already expected in immobility vs power graphs. From the graphs it is a bit difficult to convince oneself that there is linear correlation within each genotype immobility/oscillation within each genotype for recordings in the home cage or during the TST. Since immobility may be a separate cortical state, it would be helpful to assess separately the power in active and passive states of the mice. There may not be enough data point at this stage to discriminate an independent difference of immobility/oscillation between the two genotypes from a proportional relation immobility/oscillation for WT and Disc-1 mice (as suggested by the authors)*.

We performed the requested analysis and found that the power of low-frequency gamma and theta activity was reduced independent from the animal’s behavioural state (movement or immobility). This is now explicitly stated in the second paragraph of the subsection headed “Synchrony of gamma oscillations is markedly reduced in Disc1 mice”. These data are shown as new Figure 2—figure supplement 1.

The reviewer is right in assuming that there are not enough data points to discriminate independent differences in the relationship between immobility-freezing and oscillatory power between the two genotypes. To improve clarity of the presented data we explicitly state that the significant linear relationship shown in Figure 2 emerges under conditions of pooled data from controls and Disc1.

*B) Correlation oscillation/PV interneurons: the model data provide encouraging evidence for a causal role of the multiple PV defects, but this is not demonstrated. Optogenetic stimulations targeted to the PrL in Disc-1 mice might provide evidence in that direction (and even show a decrease in immobility in TST)*.

We fully agree with this important criticism and performed a new set of experiments to test whether our proposed reduction in the synaptic inhibitory output strength of PV-cells directly relates to the reduced power of network oscillations and mobility in mice. We injected recombinant adeno-associated viruses (rAAVs) encoding halorhodopsin (NpHR) and the red fluorescent protein tdTomato (rAAV-NpHR-tdT) bilaterally in the prefrontal cortices of PV-Cre mice in which Cre-combinase is only expressed in fast-spiking PV-expressing interneurons. Thus, NpHR-tdT is only expressed in Cre-recombinase positive PV-cells. We implanted optical fibers for the activation of NpHR by light (565 nm). Our aim was to silence FS-PV-interneurons and thereby reproduce the observed reduced PV-interneuron-mediated inhibition in the mPFC. However, we were unexpectedly confronted with several problems, which resulted in the time delay of our revision for which we are very sorry. The first problem was that our animals were not in the right age range and we needed to wait some weeks. The next problem was that NpHR was not as homogeneously expressed in the PV-cell population as expected from our earlier investigations in the hippocampus. Finally, we had unexpected problems with the laser for NpHR excitation. Due to the time delay, which was caused by the accumulating problems, we decided not to repeat this set of experiments for this manuscript. Thus, concerns regarding correlation and causation were addressed by careful rewriting throughout the manuscript.

*2) The statistical significance of the data needs to be re-assessed. The rationale for mixing parametric and non-parametric tests is unclear. Why is the score in the TST tested with a non-parametric test, while the FST uses a parametric test? Is the effect still significant with the Mann-Whitney test? Also, the TST-dependent cFos increase is tested on very small numbers (4 vs 3) for which the Mann-Whitney cannot detect significance anyway*.

We thank the reviewer for pointing out this issue. We indeed should have applied parametric tests to TST as well as FST measures. To re-assess statistical significance, we first tested for normality using the normaltest function of Python's stats module. Both TST and FST data followed a normal distribution. On the basis of this we applied the student *t*-test and revealed a significant difference in TST (*p*=0.015) and FST (*p*=0.049). We would like to mention that FST results would also reach significance using the Mann-Whitney test (*p*=0.029). Furthermore, we agree that the number of tested mice for cFos labelling was small and toned done our conclusions based on these results in the last paragraph of the subsection headed “Reduced inhibitory output and excitatory input signaling of fast-spiking parvalbumin-positive interneurons in the prefrontal cortex of Disc1 mice”.

*3) Many behavioral tests demonstrate no change in behavior of DISC1 mutants relative to wildtype animals. The only statistically significant finding is in the immobility scores in forced swim and tail suspension with p values of 0.039 and 0.049. Moreover, the sizes of the differences are very small, despite reaching significance. Such small differences are unlikely to be biologically significant. I would either remove or seriously downplay the behavioral analysis. The electrophysiological analysis can stand alone without the behavioral section*.

The variability in behaviour among individuals was high and resulted therefore in a moderate but significant enhancement in the mean immobility by ∼40% in the tail-suspension (TST) and by ∼18% in the forced swim test (FST) in Disc1 mice. We feel that a ∼1/3^rd^ change in spite of the high inter-individual variability in the TST is not a weak or a subtile effect. To further strengthen this point, we computed Cohen's *d*, a measure of effect size (Lakens 2013; Front Psychol 4:863). Cohen's *d* of TST and FST were 0.82 and 0.65, respectively, which corresponds to a strong and intermediate effect size, respectively. Our study shows an enormous specificity of the Disc1 mutation on the behavioural level. We would therefore like to keep the behavioural analysis in the manuscript. However, we follow the reviewers’ request and toned down the behavioural impact of the Disc1 mutation in the first paragraph of the Results section. We furthermore included Cohen's *d* measures in Table 1.

*4) The quantal analysis shown in*
Figure 4
*is, I think, a poor way to analyze the data. Model dependent quantal analysis of the type shown allows for wide variation of parameters with little difference in the goodness of fit. Nevertheless, the authors do back up the analyses with some bootstrapping models. There are very clear changes in failure rates and coefficient of variation, which are unambiguous alterations. Some additional simple analysis with Strontium to reveal the unitary current size would allow for calculation of NP*.

We used the multiple probability-compound binomial analysis (MP-CBA), developed by Silver et al. (1998; J Physiol 510:881) and Kraushaar and Jonas (2000; J Neurosci 20:5594) to estimate the number of functional release sites at FS-IN to PC synapses. Following the suggestion of the reviewer we further quantified and compared quantal IPSCs in PCs of the PrlC of Disc1 and control slices. First, we recorded mIPSCs in the presence of 0.5 µM TTX and show that they had similar amplitudes among genotypes (Disc1: 20.1 ± 1.1 pA; control: 21.9 ± 1.9 pA; *p*=0.388). These values were also similar to quantal IPSCs obtained from FS-IN to PC paired recordings in the dentate gyrus of wild type mice (18.75 ± 1.75 pA; 4 pairs) at low release probability (0.5 mM extracellular Ca^2+^) and similar to quantal IPSCs defined by MP-CBA in this study (∼28 pA). We therefore believe that quantal IPSC size defined by MP-CBA in this study is in a realistic range. Second, following the reviewers’ proposal, we newly performed paired recordings in slices of the PrlC in the presence of extracellular 5.5 mM SrCl2 (Disc1: 10.0 ± 0.8 pA in Disc1, 4 pairs; control: 12.0 ± .1.5 pA; 5 pairs; *p*=0.195). As mentioned in point 4 of our response to the reviewing editor, we noted that the IPSC amplitudes measured in SrCl2 were smaller than mIPSCs or predictions from our MT-CBA which may be caused by a direct so far unexamined effect of strontium on GABAA receptors. However, to further follow this issue, we calculated *N* for the amplitudes *A* of uIPSCs using the *q* obtained in SrCl_2_ quantal recordings according to *A*=*N***P***q*. We performed the calculation over a wide range of *P* values (0.2-1). This analysis showed that, while the resulting predicted value of *N* depends as expected on *P*, there is always a significantly smaller *N* in Disc1 compared to control pairs (*p*<0.05). These data are now included in Figure 4—figure supplement 4 and in the second paragraph of the subsection headed “Reduced inhibitory output and excitatory input signaling of fast-spiking parvalbumin-positive interneurons in the prefrontal cortex of Disc1 mice”.

*5) Since connection probability decreases as distance between soma increases for FS to Pyr cell pairs (Packer and Yuste, 2011), the authors should report the distance between soma of connected and unconnected pairs. An additional plot showing the relationship of distance between connected pairs and failure rate would also be informative*.

The axonal distance between pre- and postsynaptic cells did not differ between Disc1 and controls. In fact, we show that the average synaptic latency, which can be used as indirect measure for synaptic distance (Strüber et al. 2015; PNAS 112(4):1220-1225), was the same beween Disc1 and controls (Figure 4—figure supplement 1). This is now more clearly stated in the Results section. Non-connected pairs were not processed for morphological analysis. Therefore we cannot provide data on inter-somatic distances of non-connected simultaneously recorded cells. Furthermore, since connection probability declines with distance, as stated by the reviewers, we always aimed for inter-somatic distances of <60 µm in all paired recordings. This is now clearly stated in the Methods section under the subsection headed “In vitro electrophysiology and data analysis”. We furthermore followed the request of the reviewers and included a new graph of failure rate versus latency in Figure 4—figure supplement 1.

*6) Using anatomy (*Figure 3*) and physiology (*Figure 4*), the authors conclude that single FS terminals onto pyr cells are of the same strength but fewer in number. Does this result in a shift away from input from PV-INs onto Pyr cells, while other inhibitory inputs compensate for the loss of inhibition, or are total levels of inhibition received by pyr cells decreased? Examining mIPSCs and/or evoked IPSCs onto pyramidal cells would address this issue. This is not a necessary experiment to make the points in the paper, but would assist in constraining the models shown at the end of the paper*.

Following the suggestion of the reviewers we performed a new set of experiments and measured mIPSC frequency in the presence of 0.5 µM TTX in PCs of the PrlC of Disc1 and control mice. We observed a significant reduction in their frequency from 1.18 ± 0.23 Hz in controls to 0.74 ± 0.14 Hz in Disc1 mice (15 vs 24 cells; *p*=0.025). In contrast, the mean peak amplitude of mIPSCs was unchanged (control: 21.9 ± 1.9 pA vs Disc1: 20.1 ± 1.1 pA; *p*=0.388). We therefore found no evidence for compensatory mechanisms. These data are now included under the subsection headed “Reduced inhibitory output and excitatory input signaling of fast-spiking parvalbumin- 160 positive interneurons in the prefrontal cortex of Disc1 mice” (and in Figure 3 and Figure 4—figure supplement 4.

*7) We find sentences like* “*We find that DISC1 mice show depression-related behavioral despair” inappropriate. This is a particular interpretation of a simple increase the immobility of an animal. This is better suited for the Discussion and should be phrased as one particular interpretation of a subtle behavioral difference*.

According to this request we reformulated the sentence in the second paragraph of the Discussion as well as in the third paragraph of the Results. However, we would like to emphasize that reduced immobility during the tail suspension and forced swim test is broadly accepted as behavioural correlate of depression-related behavior in rodents and was used and also termed as depression-related behavior in various publications such as Steru et al. 1985 (Psychopharmacology 85:367-370), [30] (Nature 266:730-732) and Tye et al.; 2013 (Nature 493:537-541).